# Detecting material state changes in the nucleolus by label-free digital holographic microscopy

Christiane Zorbas [iD][1,3], Aynur Soenmez [iD][1,3], Jean Léger [iD][2], Christophe De Vleeschouwer [iD][2] &
Denis LJ Lafontaine [iD][1✉]

## Abstract

**Ribosome biogenesis is initiated in the nucleolus, a multiphase biomolecular condensate formed by liquid-liquid phase separation. The nucleolus is a powerful disease biomarker and stress biosensor whose morphology reflects function. Here we have used digital holographic microscopy (DHM), a label-free quantitative phase contrast microscopy technique, to detect nucleoli in adherent and suspension human cells. We trained convolutional neural networks to detect and quantify nucleoli automatically on DHM images. Holograms containing cell optical thickness information allowed us to define a novel index which we used to distinguish nucleoli whose material state had been modulated optogenetically by blue-light-induced protein aggregation. Nucleoli whose function had been impacted by drug treatment or depletion of ribosomal proteins could also be distinguished. We explored the potential of the technology to detect other natural and pathological condensates, such as those formed upon overexpression of a mutant form of huntingtin, ataxin-3, or TDP-43, and also other cell assemblies (lipid droplets). We conclude that DHM is a powerful tool for quantitatively characterizing nucleoli and other cell assemblies, including their material state, without any staining.**

**Keywords** Digital Holographic Microscopy; Nucleolus; Liquid-Liquid Phase Separation (LLPS); Biomolecular Condensate; Lipid Droplet
**Subject Category** Methods & Resources

## Introduction

Ribosomes are essential nanomachines responsible for protein production in cells. Ribosome biogenesis is initiated in the nucleolus, a dynamic biomolecular condensate formed by liquid-liquid phase separation (LLPS) (Lafontaine et al, 2020). The nucleolus is the most prominent membraneless organelle in the cell nucleus, its morphology reflects its role in ribosome biogenesis and other functions important for cell homeostasis (Boisvert et al, 2007;

Pederson, 1998). The nucleolus is rich in RNA and proteins that associate transiently through multivalent weak interactions to perform functions (Feric et al, 2016; Mitrea et al, 2016). The dynamic nature of the nucleolus can be explained in part by its liquid properties. These can be optogenetically tuned, with an impact on function (Zhu et al, 2019).

The nucleolus is a potent disease biomarker and stress biosensor, as its size, shape, and number per cell nucleus are markedly altered in cancer, viral infections, neurodegeneration, and ageing (Boulon et al, 2010; Derenzini et al, 2009; Salvetti and Greco, 2014; Tiku and Antebi, 2018). Despite its remarkable properties, the nucleolus remains largely underused in clinical work for lack of easy-to-implement, robust quantitative tools.

The nucleolar structure has been abundantly studied at the light and electron microscopy levels for many years (Hernandez-Verdun et al, 2010). In human cells, the nucleolus is organized in three major layers, nested like Russian dolls, and encased in a sheath of perinucleolar chromatin. The three main internal layers are the fibrillar center (FC), the dense fibrillar component (DFC), and the granular component (GC). A single GC contains multiple modules, each comprising an FC core surrounded by a DFC.

A wide range of microscopy techniques have been used in combination with dedicated staining methods to detect the nucleolus quantitatively in cultured cells and tissue biopsies. In principle, this requires either particular labeling chemistry, for example, silver nitrate-based AgNOR staining of the argyrophilic proteins which abound in the nucleolus (Bartholome et al, 2019; Ploton et al, 1986; Thelen et al, 2021), the use of specific antibodies for immunodetection, or expression of fluorescently tagged proteins for direct detection (Nicolas et al, 2016; Stamatopoulou et al, 2018; Stenstrom et al, 2020). More recently, super-resolution techniques have been applied to fixed and live samples, revealing the existence of additional nucleolar subphases (Ide et al, 2020; Yao et al, 2019). While these techniques are extremely powerful in describing the most intricate details of this fascinating biomolecular condensate, their high sophistication remains a clear obstacle to their routine use.

Digital holographic microscopy (DHM), invented by Dennis Gabor in 1948 (Gabor, 1948), is a non-invasive label-free quantitative interferometric technique that can be applied with minimal manipulation to any transparent specimen, such as fixed

[1]RNA Molecular Biology, Fonds de la Recherche Scientifique (F.R.S./FNRS), Université libre de Bruxelles (ULB), Biopark campus, B-6041 Gosselies, Belgium. [2]ICTEAM-ELEN, Fonds de la Recherche Scientifique (F.R.S./FNRS), UCLouvain, B-1348 Louvain-la-Neuve, Belgium. [3]These authors contributed equally: Christiane Zorbas, Aynur Soenmez.
✉E-mail: denis.lafontaine@ulb.be

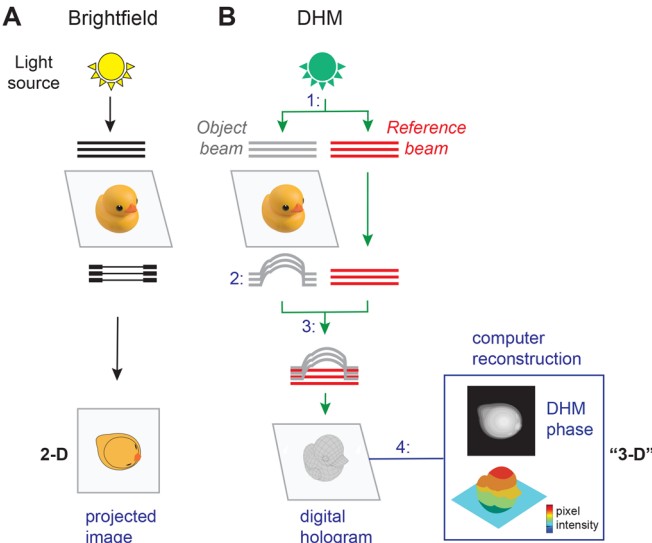

**Figure 1. Principle of digital holographic microscopy (DHM).**

(A) In brightfield microscopy, a light source illuminates an object (here illustrated as a duck), dense areas of the specimen absorb light, and the transmitted light is projected and recorded as an image. (B) In DHM, in order to generate an "interference pattern" or hologram, 1: an illumination source is split into an "object beam" (object wave light) and a "reference beam" (reference wave light), 2: the object beam passes through the sample and is subjected to a phase shift, creating the "object wavefront", 3: the "object wavefront" and the "reference wavefront" are combined to interfere and to create a hologram which is captured with a CCD camera, 4: a numerical reconstruction algorithm is used to produce a phase image from the digitally captured hologram. The phase image, which represents the optical thickness of the object at each point, can be displayed as a pseudo 3-D map using pixel intensity.

or live cells. Unlike most microscopy techniques, which record absorption and transmission of light from an object (Fig. 1A), DHM records shifts in the light wavefront with respect to a reference wavefront, producing a digital hologram from which a phase image is extracted digitally by numerical reconstruction (Fig. 1B). A key feature distinguishing DHM from optical-contrast-enhancing imaging techniques is that in DHM, the intensity value of a pixel has a direct physical meaning: it is proportional to the optical thickness (also called optical path length) of the cell, which is the physical height of the cell (or cell thickness) multiplied by its refractive index at that point (Picart, 2015; Picart and Li, 2012). Because of well-described optical artifacts, such quantitative information cannot be extracted from the images obtained by conventional brightfield microscopy or other contrast-enhancing imaging techniques such as Zernicke's' phase-contrast (PhC) microscopy or Smith and Nomarski's differential interference contrast (DIC) microscopy (see (Marquet et al, 2005) for details). DHM captures can be displayed as intuitive pseudo 3-D images resembling topographic maps, where the height is determined by the brightness of each pixel (Figs. 1 and 2A). Thus far, DHM has been exploited in material science, cell biology, and cancer studies. It has notably been used to monitor cell structure and dynamics in various biological and biomedical contexts, such as cell growth monitoring, cell dry mass estimation (Barer, 1952), drug-induced cytoskeleton dynamics (Kemper et al, 2006), neuronal growth, and metastasis progression (for a review see (Marquet et al, 2014).

DHM is also particularly well-suited for imaging liquid samples and performing in-flow analyses (Singh et al, 2017).

Here we have used DHM imaging of human cells combined with deep learning to detect and characterize the nucleolus quantitatively without any staining. The numerical parameters extracted automatically include the mean number of nucleoli per cell nucleus, the mean nucleolar area, and the mean nucleolar-to-nuclear area ratio. In addition, we have defined a novel index, the nucleolar optical thickness, demonstrating that it can be used to distinguish alterations in the material state of the nucleolus, such as those induced by opto-gelation. It was also possible to discriminate functional nucleoli from ones characterized by impaired ribosome biogenesis caused by drug treatment or ribosomal protein depletion. Lastly, we have started to explore the potential of DHM technology to detect other natural and pathological condensates (including condensates associated with neurodegeneration and stress-induced cytoplasmic granules) and other cell assemblies (including lipid droplets and mitochondria).

## Results

### Detection of the nucleolus by digital holographic microscopy

Unlike other phase-contrast-generating techniques and brightfield, digital holographic microscopy is quantitative, because, in a DHM phase image, the intensity of each pixel reflects the optical thickness of the cell, i.e. its physical height multiplied by its refractive index (Picart, 2015; Picart and Li, 2012). With this in mind, we expected optically dense cell structures, such as the nucleolus, to appear on DHM phase images as bright spots and less dense objects, such as vacuoles or the cytoplasm, as less intense areas. As shown below, this is indeed the case.

To test if DHM is suitable for detecting nucleoli, various cell lines were observed, starting with a common model: cervix carcinoma cells (HeLa). Cells were grown in a channel slide and observed directly under the microscope after brief fixation. It was also possible to observe live cells (see below). The nucleus contour was clearly identifiable in all cells (Fig. 2A, blue arrowheads). Within the nucleus, prominent structures reminiscent of nucleoli were detected (red arrows). From the quantitative information embedded in the DHM phase image, a "3-D" map was generated from pixel intensities, revealing the optical thicknesses of individual cell substructures (Fig. 2A, right). In such maps, the structures reminiscent of nucleoli appeared as sharp red peaks.

In our initial work, we used a standard digital holographic microscope with a beam path exactly as described in Fig. 1B (e.g., the phase image and 3-D display in Fig. 2A). Soon, we realized that if we wanted our method to be used in other laboratories worldwide, it would be greatly advantageous to use a "plug-in" device that would convert any inverted microscope to DHM. In the remainder of our work, we used a purposely built DHM adapter (QMOD), also referred to as "off-axis differential interferometer", directly connected to a classical CCD camera and to an inverted microscope (see Fig. EV1). In this easy-to-implement setup (see Materials and Methods), the beam path was slightly more complex than that described in Fig. 1, but the principle of quantitative interferometry was exactly the same.

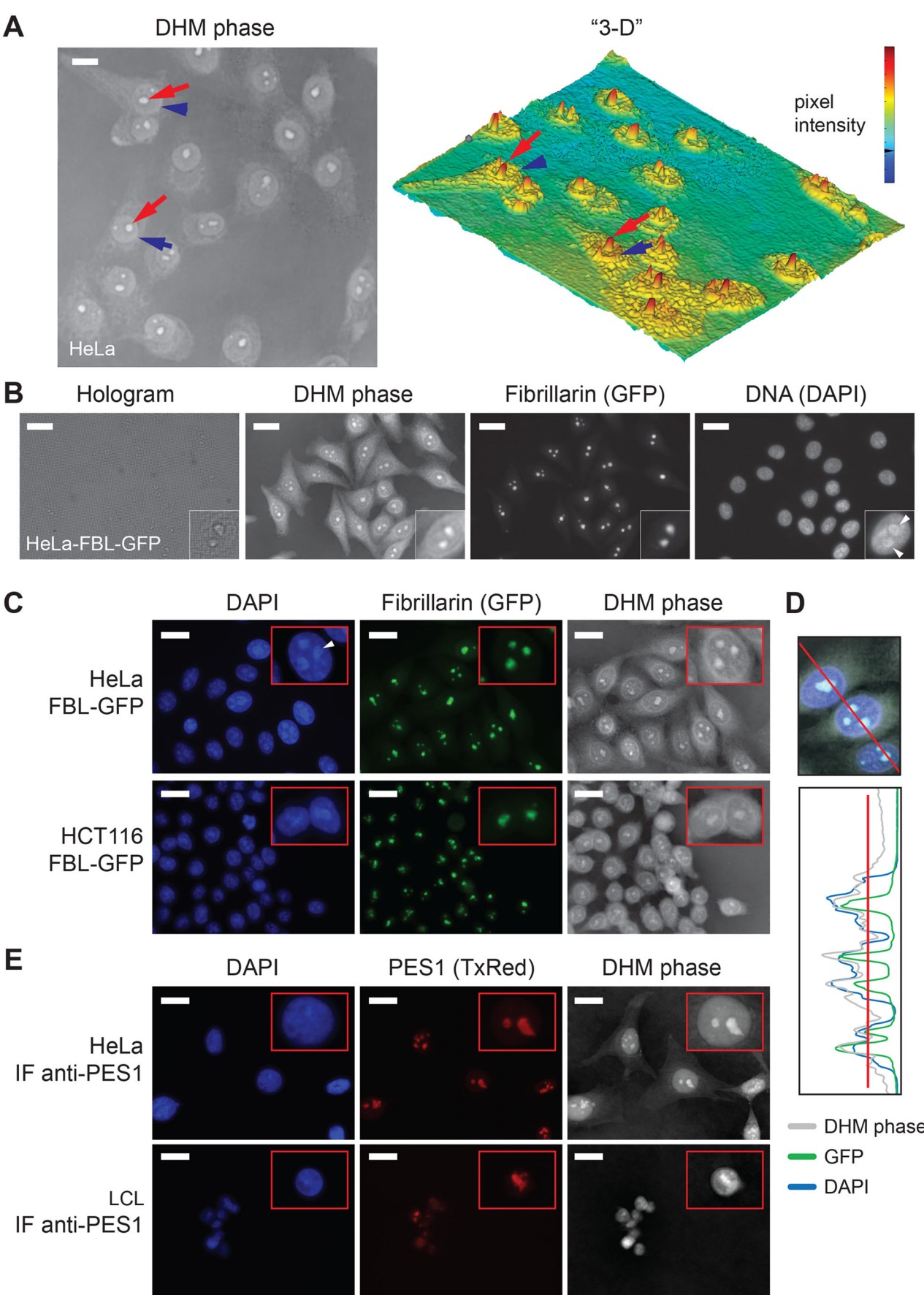

◀ **Figure 2.   The nucleolus can be detected by DHM.**

(A) (Left) HeLa cells visualized by DHM. In each cell, the contour of the nucleus is readily detectable (blue arrowhead); within the nucleus one or several intense structures are detected (red arrow); these correspond to nucleoli. Scale bar, 20 μm. (Right) Pseudo 3D map display of the image shown on the left, based on pixel intensity and representing the optical thickness. (B) HeLa cells stably expressing the green-fluorescently tagged nucleolar protein fibrillarin (HeLa-FBL-GFP) observed by correlative DHM-fluorescence microscopy. Cells were stained with DAPI, which labels the DNA-rich nucleoplasm. The hologram, DHM phase, green fluorescence (GFP), and DAPI signals are shown. Insets, magnification of an individual cell nucleus illustrating the perinucleolar chromatin ring (arrowheads in the DAPI panel). Scale bar, 20 μm. (C) HeLa-FBL-GFP and HCT116-FBL-GFP cells were observed by correlative DHM-fluorescence microscopy. The DAPI (nucleoplasm), green fluorescence (nucleoli), and DHM phase signals are shown. Insets, magnification of individual cell nuclei. Scale bar, 20 μm. The white arrowhead points to the perinucleolar chromatin ring visible in the DAPI channel. (D) Quantification traces of the DHM phase (in gray), green fluorescence (in green), and DAPI (in blue) signals. (E) Immunodetection of the nucleolar protein PES1 in HeLa and LCL cells visualized by correlative DHM-fluorescence microscopy. PES1 was imaged in red (Texas red, TxRed). The DAPI, green fluorescence, and DHM phase signals are shown. Insets, magnification of an individual cell nucleus. Scale bar, 20 μm. Source data are available online for this figure.

A major advantage of the QMOD is that DHM can be readily combined with fluorescence imaging to perform correlative DHM-fluorescence microscopy. This is what we did to ascertain that the prominent structures detected in the nucleus were indeed nucleoli (Fig. 2B–E). Initially, we used HeLa cells stably expressing the nucleolar protein fibrillarin fused to a green fluorescent tag (HeLa-FBL-GFP). The nucleolus consists of three main layers nested like Russian dolls (Thiry and Lafontaine, 2005), and fibrillarin marks the middle layer or dense fibrillar component. Cells were stained with DAPI, which labels the DNA-rich nucleoplasm. Comparing the fluorescence signal (fibrillarin, GFP) with the DHM phase made it obvious that the prominent nuclear substructures detected by DHM were nucleoli. The nucleolus is lined by a layer of condensed chromatin, the so-called perinucleolar chromatin forming a distinctive DNA "ring" around the nucleolus. This ring was visible in the DAPI images (see arrowheads in Fig. 2B, DAPI inset). The presence of DNA rings circling the prominent nuclear foci observed in the DHM phase images further confirmed their identity as nucleoli. Quantification of the fluorescence and DHM signals demonstrated an excellent overlap between phase intensity and GFP peaks, formally confirming colocalization (Fig. 2D).

To extend our observations to other cells, we used a colon carcinoma cell line (HCT116), also stably expressing a FBL-GFP construct (Fig. 2C). In these cells, the DHM phase images again revealed prominent signals in the nucleus, confirmed to be nucleoli on the basis of colocalization with fibrillarin and counterstaining with DAPI (Fig. 2C).

In addition to using fluorescently tagged nucleolar proteins, we detected endogenous proteins by indirect immunofluorescence with specific antibodies. In this experiment, we used both adherent cells (HeLa) and suspension cells (lymphoblastoid cells, LCL) and chose to detect the nucleolar protein PES1 (pescadillo ribosomal biogenesis factor 1, Fig. 2E). PES1 labels the cortical layer of the nucleolus or granular component. In both cases, we observed excellent colocalization between the DHM phase and the fluorescence signal (Fig. 2E). This confirmed that the prominent nuclear foci detected by DHM were nucleoli.

To expand the scope of our observations, we analyzed four additional adherent cell lines: another type of cervix carcinoma cells (SiHa), bone cancer cells (U2OS), breast cancer cells (MCF7), and lung cancer cells (A549) (Fig. EV2A). All cells displayed prominent nucleolar signals, similar to those observed in HeLa and HCT116 cells. Since the detection of nucleoli was a bit more difficult in suspension cells than in adherent cells, we repeated the original observation with a second LCL line and two more suspension cell lines: Jurkat and K562 (Fig. EV2B). Jurkat cells are immortalized T lymphocytes, and K562 cells are immortalized myelogenous leukemia cells. We conclude that the detection of nucleoli is also readily achievable in diverse suspension cells. Thus, DHM allows robust stain-free detection of the nucleolus in adherent and suspension cells of various tissue origins.

## Nucleolar structure alterations are detectable by digital holographic microscopy

Morphological alterations of the nucleolus are associated with diverse pathological conditions ranging from cancer, viral infection, and neurodegeneration to various types of cell stress and even ageing (Boulon et al, 2010; Derenzini et al, 2009; Tiku and Antebi, 2018).

We wondered if such nucleolar alterations could be detected by DHM. To test this possibility, we induced nucleolar alterations by treating cells with several drugs known to disrupt the nucleolus (Burger et al, 2010) or by depleting them of factors important for nucleolar structure maintenance (Nicolas et al, 2016; Stamatopoulou et al, 2018).

In these experiments we used HeLa cells stably expressing an FBL-GFP construct to visualize the dense fibrillar component of the nucleolus. After drug treatment, we performed immunostaining with an anti-PES1 antibody to additionally visualize the granular component.

First, we treated cells with either DRB (5,6-dichloro-1-β-D-ribofuranosyl-1H-benzimidazole), roscovitine (ROS), actinomycin-D (ActD), or CX-5461 (Fig. 3A). DRB and roscovitine, which are both Cdk-activated kinase inhibitors, are well known to unfold the nucleolus into beaded strands called nucleolar necklaces (Fig. 3A and see Lafontaine et al, 2020; Shav-Tal et al, 2005). They belong to a group of inhibitors of kinases important for transcriptional elongation (Bensaude, 2011). DRB targets CDK9 in P-TEFb and roscovitine inhibits a cdc2-cyclin B kinase that normally keeps RNA Pol I repressed during mitosis (Sirri et al, 2000). Actinomycin-D and CX-5461, on the other hand, are both RNA polymerase I inhibitors, and both of them clearly cause segregation of nucleolar components, which become juxtaposed in so-called "nucleolar caps", rather than remaining nested (Fig. 3A, see the white arrow Lafontaine et al, 2020). ActD is a DNA intercalator which preferentially targets GC-rich sequences and which, at the concentration used here, inhibits only Pol I (although at a higher dosage, it can inhibit any polymerases). CX-5461 reduces the binding of SL1 pre-initiation complex and RNA polymerase I complex to rDNA promoters (Bywater et al, 2012; Drygin et al, 2011) and stabilizes G-quadruplexes abundant in rDNA (Xu et al, 2017).

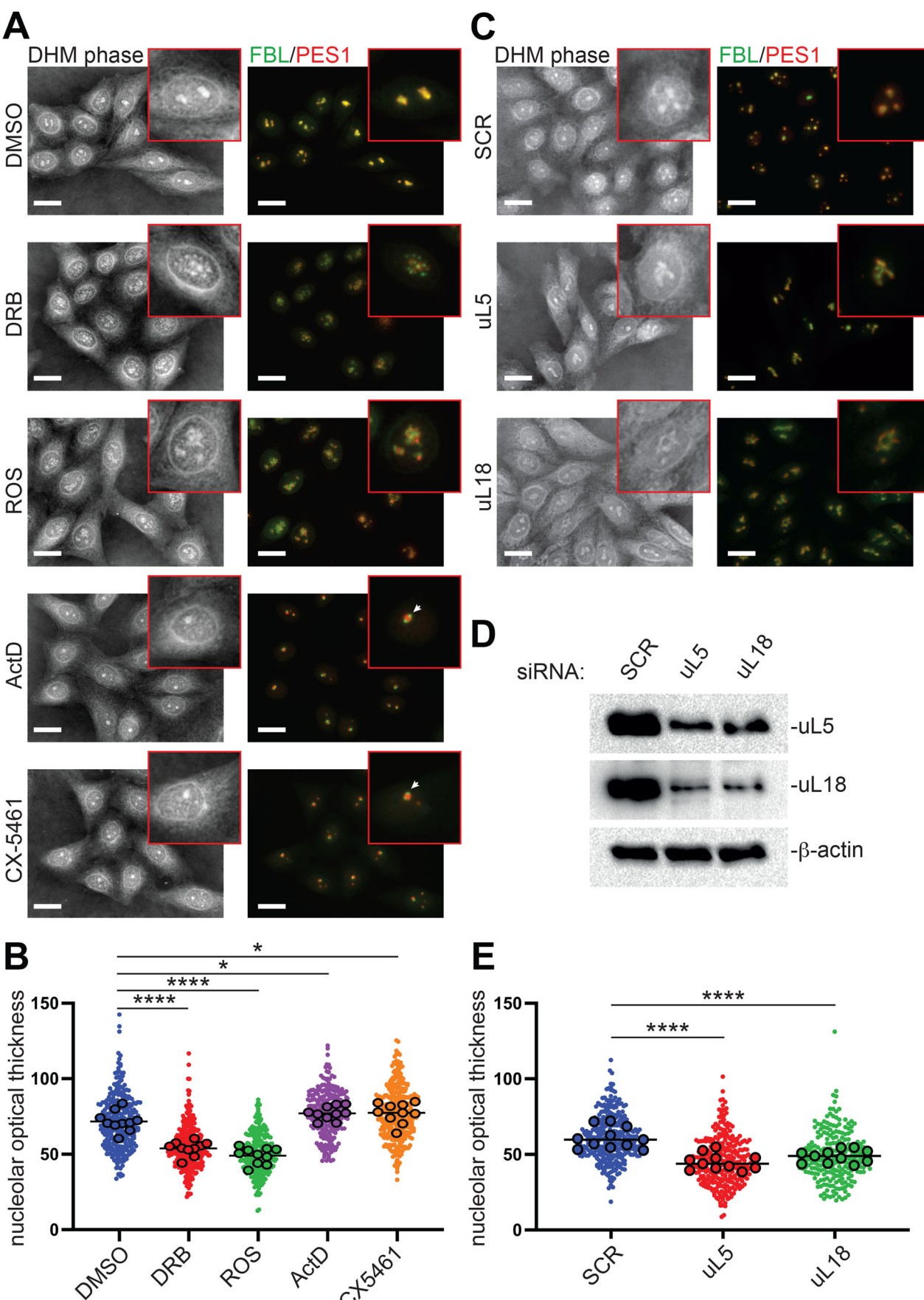

◀    **Figure 3. Nucleolar alterations can be detected by DHM.**

Diverse nucleolar alterations were induced in HeLa-FBL-GFP cells, observed by correlative DHM-fluorescence microscopy, and quantified. (A) Drug-induced nucleolar alterations. Cells were treated for 2 h with DRB (0.1 mM), roscovitine (ROS, 50 µM), actinomycin-D (ActD, 160 nM), CX-5461 (5 µM), or the vehicle control (DMSO 0.1%). White arrows point towards nucleolar caps (ActD and CX-5461 pictures). Scale bar, 20 µm. (B) Quantification of the nucleolar optical thickness reveals a significant reduction in cells treated with DRB or ROS and an increase upon treatment with ActD or CX-5461, by comparison with a DMSO control. Number of independent experiments, $n = 10$. The mean of each independent experiment is represented by a black circle, the mean of the means by a black line, and the individual nucleoli are counted by colored dots. The total numbers of nucleoli counted were 290, 256, 234, 301, and 287 for DRB, ROS, ActD, CX-5461, and DMSO, respectively. Data were analyzed by one-way ANOVA ($p < 0.0001$) and by uncorrected Fisher's LSD (DMSO vs. DRB, $p < 0.0001$; DMSO vs. ROS, $p < 0.0001$; DMSO vs. ActD, $p = 0.038$; DMSO vs. CX-5461, $p = 0.030$). *$p ≤ 0.05$; ****$p ≤ 0.0001$. (C) Nucleolar alterations induced by ribosomal protein knockdown. HeLa-FBL-GFP cells were treated for three days with a siRNA (10 nM) specific to the mRNA encoding ribosomal protein uL5 or uL18 or with a non-targeting scramble control (SCR). Scale bar, 20 µm. (D) Western blot assessment of uL5 and uL18 depletion after siRNA-mediated knockdown. ß-actin was used as a loading control. Note that uL18 is reduced upon uL5 knockdown and vice versa. (E) Knockdown of ribosomal protein uL5 or uL18 leads to a significant reduction in nucleolar optical thickness as compared to the SCR control. Number of independent experiments, $n = 11$. The mean of each independent experiment is represented by a black circle, the mean of the means by a black line, and the individual nucleoli are counted by colored dots. The total numbers of nucleoli counted were 311, 235, and 332 for uL5, uL18, and the SCR control, respectively. Data were analyzed by one-way ANOVA ($p < 0.0001$) and by uncorrected Fisher's LSD (SCR vs. uL5, $p < 0.0001$; SCR vs. uL18, $p < 0.0001$). ****$p ≤ 0.0001$. Source data are available online for this figure.

Correlative DHM-fluorescence microscopy (with detection of fibrillarin and PES1) confirmed that each treatment affects nucleolar structure deeply, as previously described in the literature. Comparing the fluorescence and DHM phase signals revealed that each alteration was readily detectable by DHM (Fig. 3A). Considering the absence of staining in the DHM phase signal, the definition of nucleolar granules was truly exceptional (see e.g., DRB images).

We additionally altered nucleolar morphology by removing factors important for its structure. We concentrated on two proteins of the large ribosomal subunit, uL5 (formerly RPL11) and uL18 (RPL5), whose depletion has been shown to have a major impact on the nucleolus (Nicolas et al, 2016). Cells depleted of either protein showed obvious nucleolar alterations in both the fluorescence mode (with FBL and PES1 detection) and the DHM mode (without staining) (Fig. 3C).

To ascertain the levels of uL5 and uL18 depletion, western blotting was performed (Fig. 3D). This confirmed the effective depletion of each protein and further revealed that the metabolic stability of each depends on the presence of the other (i.e., they codeplete). This is particularly interesting, considering that uL5 and uL18 are part of the same trimeric complex, together with the 5S rRNA. This complex is important in regulating the homeostasis of the anti-tumor protein p53 in a process known as nucleolar surveillance (Nicolas et al, 2016).

In conclusion, DHM can readily detect fine alterations of nucleolar structure induced by pharmacological treatment or by depletion of ribosomal proteins important for nucleolar structure maintenance.

The nucleolus is visible only during the interphase, being known to disassemble at the onset and reassemble at the end of mitosis (Lafontaine et al, 2020). We wanted to know if nucleolar disassembly and reassembly could be monitored by DHM. We thus performed live-cell imaging using DHM and concluded that it can (Movie EV1).

As discussed above, nucleolar alterations induced by drug treatments can be visualized efficiently by DHM on fixed cells (Fig. 3). To see if such drug-induced alterations could also be monitored dynamically in live cells, we seeded HeLa-FBL-GFP cells into channel slides, added roscovitine, and imaged live cells every 30 s for 3 h. The drug-induced nucleolar alterations could indeed be followed in live cells by DHM, since the changes observed in the GFP channel were also obvious in the phase signal (Movie EV2). Although not directly the subject of our study, another interesting observation emerged from the movies: we saw numerous cytoplasmic granules, exploring rapidly the cytoplasmic space. Co-staining of cells with MitoTracker indicated that these are mitochondria (Movie EV3).

## Quantitative analysis of the nucleolus detected by DHM, using deep learning

Having shown that the nucleolus can be detected by DHM, we next sought to use the technique to extract types of quantitative information that would normally require specific staining. Typically, numerical parameters such as the mean number of nucleoli per cell nucleus, the mean nucleolar area, and the mean nucleolar-to-nuclear area ratio. These parameters are classically used in clinical biology (Derenzini et al, 2009; Drygin et al, 2010).

A database of seventy-five fields of view was generated, each comprising about fifteen HeLa cells expressing fluorescently tagged fibrillarin. For each field of view, three images were captured: GFP, DAPI, and the DHM phase. This database was used to establish a fluorescent thresholding method for detecting nuclei and nucleoli and to train U-net convolutional neural networks. Importantly, all segmentation procedures were carefully benchmarked manually.

First, by means of fluorescence only, the nucleus and nucleoli of each cell were segmented, respectively, by thresholding the DAPI and GFP signals (Fig. 4A; Appendix Figs. S1–S3). For the detection of the nucleus, a simple thresholding method was sufficient, while for counting the nucleoli and establishing the nucleolar area, a more sophisticated thresholding method was required. This involved detecting local maxima, followed by "region growing" (see Materials and Methods). During this thresholding, most cells undergoing mitosis (during which the nucleolus is disassembled, see (Lafontaine et al, 2020)) and multi-nucleated or incomplete cells (image edges) were filtered out. Importantly, only structures whose GFP signal was contained within a DAPI signal were considered to be nucleoli. The nucleolar parameters extracted are presented in Fig. 4B. Altogether, 2882 nucleoli from 1206 cells were analyzed. This led to the conclusion that the mean number of nucleoli per HeLa cell was 2.39, the mean nucleolar area was 23.15 µm², and the mean nucleolar-to-nuclear area was 0.14.

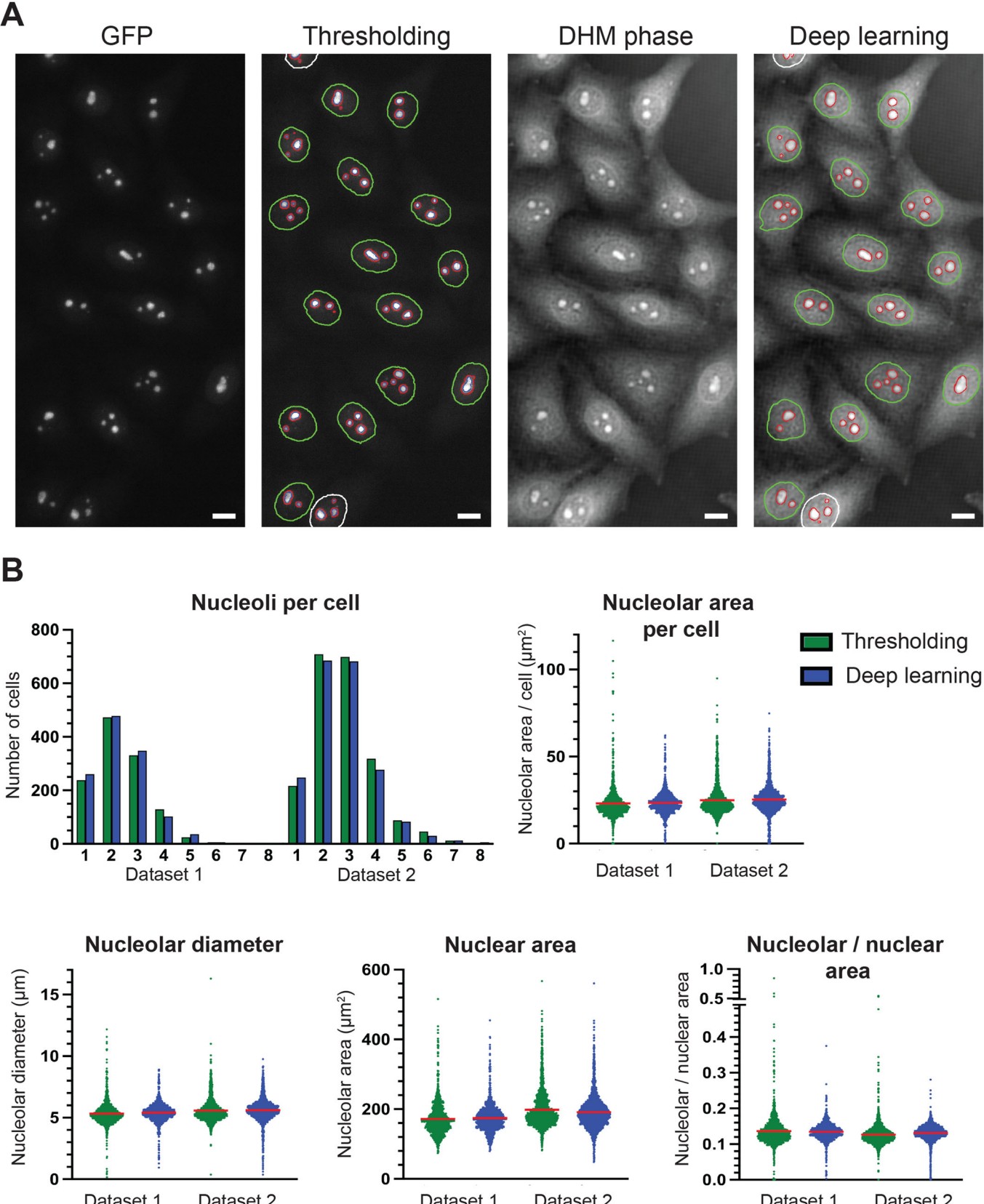

**A**

GFP | Thresholding | DHM phase | Deep learning

**B**

**Nucleoli per cell**

**Nucleolar area per cell**

Thresholding
Deep learning

**Nucleolar diameter**

**Nuclear area**

**Nucleolar / nuclear area**

**Figure 4. Nucleolar parameter analysis.**

(A) Illustration of image segmentation of HeLa cells stably expressing fibrillarin-GFP. The GFP channel was segmented by thresholding and the DHM phase by deep learning. Scale bar 10 μm. Thresholding: red, automatic segmentation of nucleoli on GFP images, for computing the nucleolar area; blue: automatic segmentation of nucleoli, for counting; green: automatic segmentation of the nuclei by thresholding. Deep learning: red, automatic segmentation of nucleoli (for counting and area computation); green, automatic segmentation of the nuclei. (B) Data analysis. The parameters were extracted automatically by thresholding (fluorescence) or deep learning (DHM phase). To benchmark the robustness of our approach, data captured more than two years apart by two different scientists were compared (Dataset 1, 75-image set; Dataset 2, 50-image set). For all parameters, it can be seen that the results obtained by thresholding or deep learning are highly consistent. For the number of nucleoli per cell, the distribution of cells with 1–8 nucleoli is shown for both datasets. For the other parameters, all individual data points are shown (up to several thousands). The red bar represents the mean. Source data are available online for this figure.

Interestingly, the mean nucleolar area was reasonably well conserved in nuclei containing up to four nucleoli (Appendix Fig. S2), after which it gradually increased. This was as expected if small nucleoli coalesce into larger ones in a liquid-liquid-like fashion, in agreement with the LLPS model of nucleolar assembly. If one views the nucleolus as a sphere, the projected areas of multiple small spheres cover a larger area than the projection of fewer large spheres. Assuming that a constant volume V is divided into N identical spheres, the radius of each sphere becomes proportional to $(V/N)^{1/3}$. Hence, the surface projected by N spheres is proportional to N times $(V/N)^{2/3}$, which is proportional to $N^{1/3}$ (Appendix Fig. S2A). Also note that the automatic counting of nucleoli on fluorescence images was carefully benchmarked with manual annotations and assessed with confusion matrices as well as sensitivity, specificity, and precision scores (Appendix Fig. S2B,C). The fraction of cells whose nucleoli were accurately counted (i.e., the precision) was superior to 83% for cells displaying one to four nucleoli. The sensitivity and specificity were >78% and >94%, respectively, when 1–4 nucleoli were counted. The most frequent errors were counting one too many nucleoli or missing one. The count was less accurate when five or more nucleoli were counted, but this situation was hardly ever encountered in cells (Appendix Fig. S4B).

The next step was to identify the nucleus and nucleoli in cells directly on the DHM phase images, without using fluorescence signals. To achieve this, two 2-D U-net convolutional neural networks were trained (Falk et al, 2019), one using as input the segmented DAPI signal, the other using the segmented GFP signal (Fig. 4; Appendix Figs. S1B,C and S3B). The database of images was divided into three groups of twenty-five fields of view each. Two groups were combined for training and the third was used for testing. The operation was reiterated twice until each group had been used once for testing. All seventy-five fields of view were then used for automatic extraction of numerical features. As an illustration, a representative GFP image was segmented by thresholding and the corresponding DHM phase by deep learning, with nearly identical results (Fig. 4A; Appendix Fig. S3). The data show that the numerical parameters extracted directly from the DHM phase images by deep learning were highly consistent with those obtained by thresholding the GFP images (Fig. 4B). The histograms representing the number of nucleoli per cell nucleus computed from the fluorescence images and those computed from the DHM phase images are nearly identical (Appendix Fig. S2). The precision, sensitivity, and specificity computed from the confusion matrix were respectively >64%, >64%, and >80%, when one, two, or three nucleoli were counted (Appendix Fig. S2C).

Once established this new method for extracting quantitative nucleolar parameters, it was important to test its robustness. An independent dataset based on 50 new images was produced. The two datasets were acquired by independent scientists more than two years apart. The comparative analysis revealed high consistency of the extracted parameters (Fig. 4B). The differences observed may reflect marginal metabolic fluctuations associated with cell passage number, medium batch, etc.

## Changes in the material state of the nucleolus can be detected by DHM

The above experiments establish that the nucleolus is readily detectable, without any staining, by DHM and that parameters classically associated with nucleolar biology can be extracted manually or automatically by deep learning.

With the recent increased interest of cell biologists in soft matter research and biophysics, it has become clear that the material state of a cell plays an important role in homeostasis. As DHM phase images contain unique quantitative information (see Introduction), we were keen to define a novel index liable to characterize the material state and to prove useful in the booming field of biomolecular condensate research.

To test the idea that DHM measurements might offer a powerful novel means of assessing material states of the nucleolus, we designed an experiment where we converted the nucleolus from a liquid to a gel, having shown previously that it is possible to modulate locally the nucleolar material state by use of optogenetics, with consequences on nucleolar function (Zhu et al, 2019).

Briefly, we engineered a Cry2olig tag-containing nucleolar construct and expressed it directly from the genome of a HEK293 cell. The Cry2olig tag is known to self-polymerize upon exposure to blue light (488 nm), thus leading to protein aggregation and a change in material state (Fig. 5A,B) (Taslimi et al, 2014; Zhu et al, 2019). To ascertain nucleolar targeting and efficient mixing of the Cry2olig fusion protein with the nucleolar phase, a nucleolar localization signal (NoLS) was inserted into the construct in addition to an intrinsically disordered region (IDR). Additionally, a mCherry tag was used to monitor subcellular distribution by fluorescence microscopy and protein mobility by fluorescence recovery after photobleaching (FRAP).

To test for possible dose-response effects, we choose two distinct blue light (BL) illumination conditions: medium- and high-intensity exposure. We used fluorescence recovery after photobleaching (FRAP) to assess the level of gelation of the nucleolus. Upon exposure to blue light, the mobility of the Cry2olig protein construct decreased, as judged by slowed FRAP curves (Fig. 5C,D). This confirms opto-gelation of the nucleolus. We observed a correlation between the level of blue light exposure and nucleolar gelation: the more intense the exposure, the slower the fluorescence recovery after photobleaching.

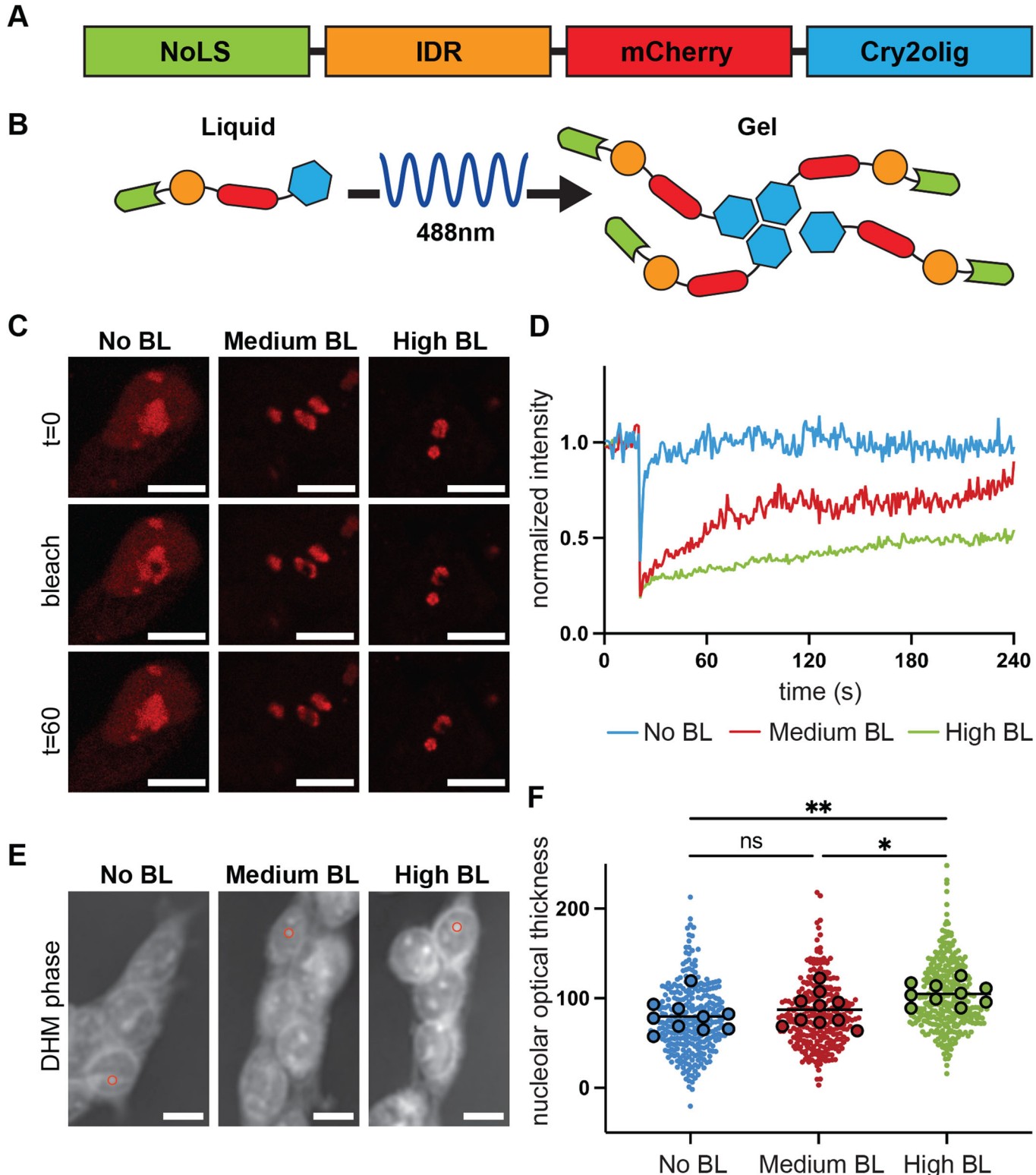

We defined a novel index, the nucleolar optical thickness, as the mean optical path length (OPL) of a 12-pixel-sized disc area centered on the nucleolus in DHM phase images, from which we subtracted the mean OPL of a similar-sized area in the adjacent nucleoplasm in order to take the background into account (see Materials and Methods for details).

Using this method and calculating the index for ~30 nucleoli in each condition in ten independent experiments, we measured general means of $78.19 \pm 17.72$, $85.58 \pm 18.66$, and $103.2 \pm 11.94$ respectively, for control, medium blue light intensity, and high blue light intensity conditions (Fig. 5E,F; Table EV1). Thus, we observed a gradual increase in nucleolar optical thickness concomitant with

**Figure 5. Nucleolar material state alterations can be detected by DHM.**

(A) Structure of the Opto-tag construct used. NoLS nucleolar localization signal, IDR intrinsically disordered region, mCherry fluorescent tag for microscopy, Cry2olig self-polymerization tag activated upon blue light exposure (see Materials and Methods for details). This construct was stably integrated into HEK293 cells. (B) The rationale of material state change: upon exposure to blue light, the opto-tag construct self-polymerizes, turning the nucleolus from a liquid to a gel. (C, D) Protein aggregation was induced upon blue light exposure to promote a material state change (gelation). To test dose responsiveness, two blue light (BL) intensities were used (medium, 8 V and high, 9.5 V). Material state change was established by fluorescence recovery after photobleaching (FRAP) in cells exposed to medium- or high-intensity blue light or not exposed to blue light (No BL = 0 V). Time scale (t), seconds. Panel (C) shows representative examples of photobleached cells. Panel (D) shows the matching FRAP curves. Note the reduction in fluorescence recovery with increasing intensity of blue light exposure. Each FRAP curve is the mean of n independent experiments with $n = 6$ (No BL), $n = 4$ (Medium BL), $n = 13$ (High BL). Scale bar 10 μm. (E) DHM phase images of cells prior to and after exposure to blue light at different intensities. Red circle, a 12-pixel disc area used for quantification. Scale bar 10 μm. (F) Quantification of optical thickness in DHM phase images reveals a gradual increase upon blue light exposure. Exposure to medium-intensity blue light increases nucleolar optical thickness by ~9%, while exposure to high-intensity blue light increases the nucleolar optical thickness by ~32%. Number of independent experiments, $n = 10$. The mean of each independent experiment is represented by a black circle, the mean of the means by a black line, and the individual nucleoli are counted by colored dots. The total numbers of nucleoli counted were 300, 301, and 335 for no BL, medium BL, and high BL, respectively. Data were analyzed by one-way ANOVA ($p = 0.0062$) and by uncorrected Fisher's LSD (no BL vs. medium BL, $p = 0.322$; no BL vs. high BL, $p = 0.002$; medium BL vs. high BL, $p = 0.023$). ns, $p > 0.05$; *$p \le 0.05$; **$p \le 0.01$. Source data are available online for this figure.

gelation, reaching ~32% between the control and high blue light intensity conditions.

In conclusion, DHM can effectively distinguish nucleolar material states in a dose-response manner.

Having demonstrated that differential OPL assessment can distinguish distinct nucleolar material states, we wondered if it could also deconvolute the alterations caused by drug treatment or factor depletion (See above Fig. 3). Using the same OPL measurement method as in the optogenetic experiment, we show that it could (Fig. 3B,E).

Not only did the OPL values of the "treatment conditions" always differ from those of the controls (a control with the drug vehicle DMSO and one with the non-targeting scramble silencer SCR, Fig. 3B,E; Table EV2), but quite remarkably, they allowed grouping the drugs in agreement with their known effects on nucleolar structure. On the one hand, DRB and roscovitine, which both leads to the formation of small nucleolar aggregates, caused an OPL decrease. On the other hand, actinomycin-D and CX-5461, which both inhibit Pol I and lead to the formation of nucleolar caps, caused an OPL increase (see Fig. 3B; Table EV2). Lastly, we found depletion of uL5 or depletion of uL18, which affect the same assembly step of the large ribosomal subunit (formation of the central protuberance) with similar consequences on nucleolar morphology (Nicolas et al, 2016), to cause a similar OPL reduction (Fig. 3E; Table EV2).

Having shown that DHM successfully detects nucleolar material state changes in the context of opto-gelation and of inhibition of function caused by drug treatment or factor depletion, we wanted to see how a drastic treatment known to affect mostly the cytoplasm might affect the nucleolus.

We chose to treat cells with latrunculin A, a drug known for its actin-depolymerizing effects leading to cytoskeleton collapse (Spector et al, 1983). We measured optical thickness as described above. In the cytoplasm, we observed a mild but noticeable reduction, in line with the loss of structural integrity (Fig. EV3; Table EV3). For the nucleolus, in contrast, we observed a marked increase, accompanied by increased circularity and a decreased area (Fig. EV3; Table EV3). We suggest the observed differences reflect the loss of cytoskeleton caused by latrunculin A treatment, accompanied by an observed "rounding" of the nucleus resulting in increased pressure on the nucleolus.

In conclusion, DHM is also suitable for monitoring drastic effects such as those inflicted on cells upon actin depolymerization.

## Exploring the potential of DHM to detect other cell assemblies in the nucleus and cytoplasm

The nucleolus is a very large nuclear condensate. With a view to expanding the applicability of DHM in the condensate field, we examined whether smaller condensates inside and outside the nucleus might also be detected by this technique.

We first focused on pathological condensates observed in cases of neurodegeneration and known to form upon expression of a mutant form of a protein. We chose to express constructs encoding mutant forms of huntingtin, ataxin-3, or TDP-43, as such mutations have been shown to cause the pathological formation of condensates. In each case, a GFP fusion was used for direct visualization.

The expressed mutant forms of huntingtin and ataxin-3 both carry a pathogenic poly-glutamine extension. That of the mutant huntingtin, associated with Huntington's disease, is 72 amino acids long (Narain et al, 1999). That of the altered ataxin-3, associated with spinocerebellar ataxia type 3 (also known as Machado-Joseph disease), is 84 amino acids long (Chai et al, 2002). The altered TDP-43 is a C-terminal fragment of the native protein, associated with amyotrophic lateral sclerosis (ALS), frontotemporal dementia (FTD), and Alzheimer's disease (AD) (Yang et al, 2010).

As expected, expression of the mutant form of huntingtin, ataxin-3, or TDP-43 led to the formation of remarkable foci clearly identifiable in the GFP channel (Fig. 6) in either the nucleus (red arrowheads) or the cytoplasm (white arrowheads). These foci, which were either similar in size to the nucleolus or several folds smaller, were all well identifiable in the DHM phase without any staining (see overlay and zoom panels, Fig. 6).

We also wondered if natural condensates other than the nucleolus might be monitored. We focused on cytoplasmic stress granules (SGs), as these can be induced by arsenite-mediated translational inhibition (Fig. EV4). Using a cell line stably expressing the RasGAP-associated endoribonuclease G3BP (a marker of SGs, see Tourriere et al, 2023) in fusion with a green fluorescent epitope, we easily detected cytoplasmic foci upon treatment with arsenite. On close inspection, we realized that at least some SGs could also be detected on DHM phase images (Fig. EV4).

Lastly, we turned our attention to another type of cell assembly, this time membrane-bound: lipid droplets. These are central to cell

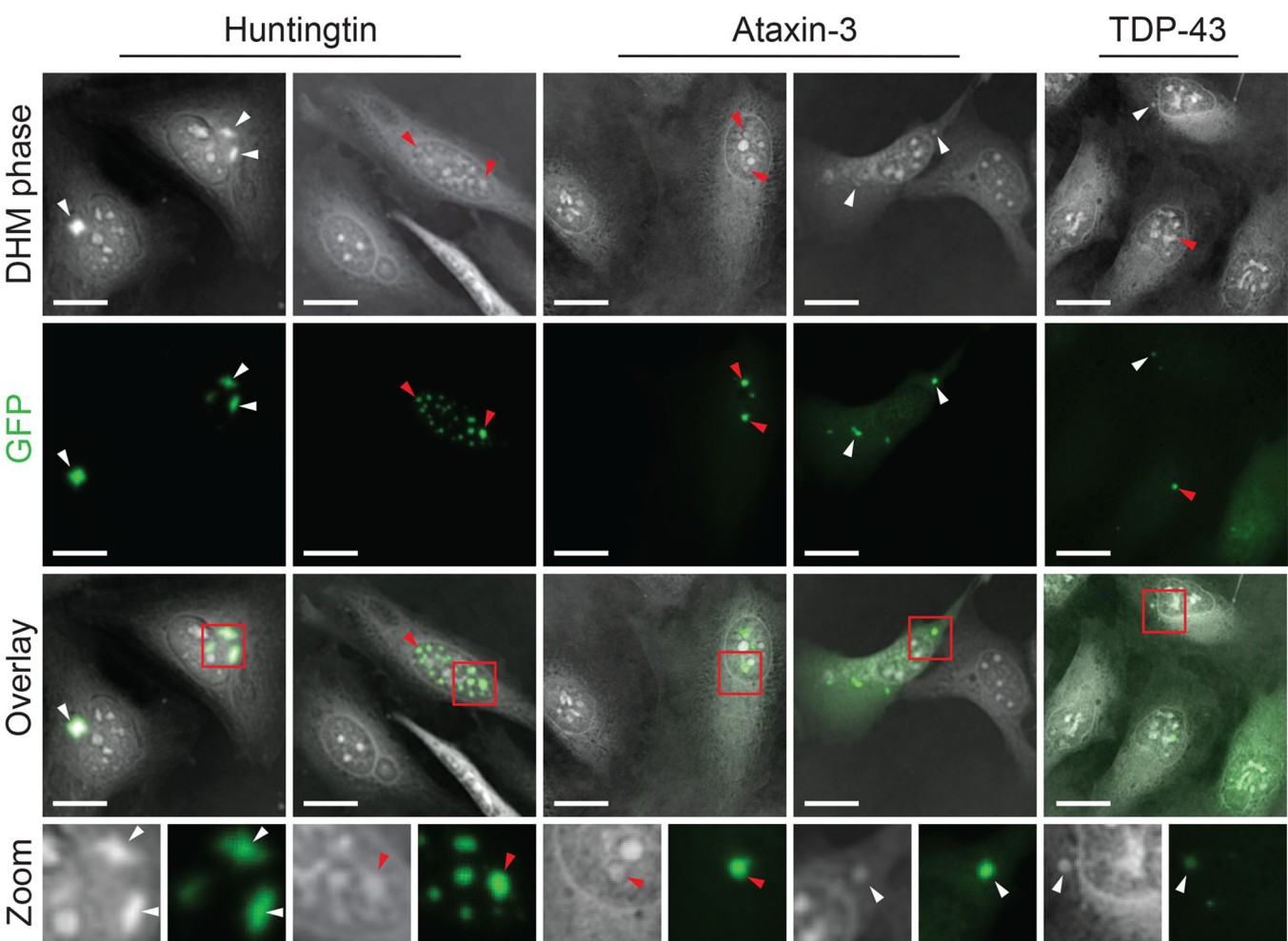

**Figure 6. Disease condensates can be detected by DHM.**

DHM successfully detects disease protein aggregates. Cells were transfected with plasmids expressing GFP-tagged mutant forms of proteins known to form aggregates in neurodegenerative diseases. Condensates were detected in both the nucleus (red arrowheads) and cytoplasm (white arrowheads). HeLa cells were transfected with the huntingtin or TDP-43 construct. U2OS cells were transfected with the ataxin-3 construct. Arrowheads highlight representative condensates visible in both the GFP and DHM channels. Scale bar 20 μm. Source data are available online for this figure.

energy metabolism and consist of "oil-in-water" emulsion droplets covered by a phospholipid monolayer. Our prime motivation stemmed from the important connections that are emerging between membrane-bound organelles and membraneless condensates, with (1) membranes sometimes playing important roles in initiating condensate formation and in regulating condensate size and/or material properties, (2) condensates forming inside membrane-delimited organelles (Kusumaatmaja et al, 2021; Snead and Gladfelter, 2019; Zhao and Zhang, 2020), (3) lipids accumulating inside condensates (Dumelie et al, 2023), and (4) quite remarkably, condensates (stress granules) "plugging" leaks in ruptured membrane-bound vesicles (Bussi et al, 2023).

In fact, lipid droplets themselves share properties with condensates, as their interior consisting mostly of neutral lipids (triacylglycerols and/or sterol esters) has to reach a critical concentration during biogenesis in order to phase separate and condense in a process termed nucleation (Santinho et al, 2020). In addition, lipid droplets are mostly cytoplasmic (the cytoplasm

being a space we have only started to probe by DHM), their formation can be induced in a controlled fashion in cell differentiation protocols, they span a wide range of sizes, and they can fuse into sizeable structures.

For all these reasons, we chose to differentiate human adipose-derived stem cells into adipocytes over a two-week period and to monitor the process by DHM. First, we followed lipid droplet formation by classic Oil Red O staining (ORO); as a reference, the nucleus was stained with DAPI (Fig. 7A). Lipid droplets were apparent by day 6 and gradually became larger and more abundant over time, up to day 15.

ORO staining is not compatible with DHM phase acquisition, so in a second series of experiments, we visualized lipid droplets by counterstaining them with an antibody specific to perilipin 1 (PLIN1), which labels the contour of lipid droplets (Fig. 7B, in green).

As can be seen in Fig. 7B, DHM turned out to be an extremely powerful tool for visualizing lipid droplet formation, as tiny foci

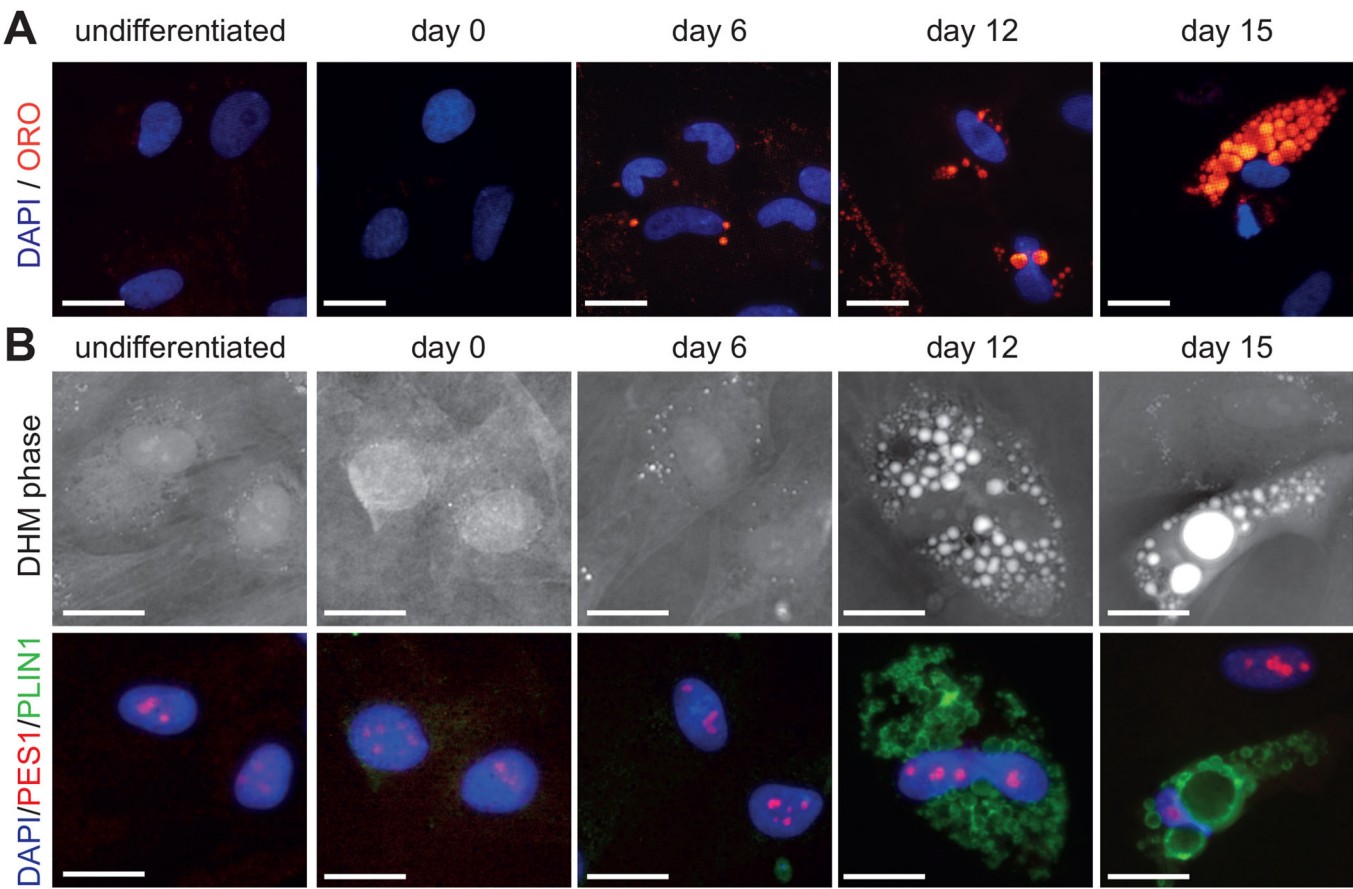

**Figure 7. Lipid droplets can be detected by DHM.**

(A) Differentiation of proliferating adipose-derived stem cells to adipocytes over a 2-week time course. Cells were inspected prior to differentiation (undifferentiated) upon initiation of differentiation (day 0), and 6-, 12-, and 15- days after initiation of differentiation. Lipid droplets, stained with Oil Red O (ORO), were first visible on day 6 and showed increased size and number on days 12 and 15. DAPI was used to stain the nucleoplasm. Scale bar, 20 μm. (B) Correlative DHM-fluorescence microscopy demonstrating that DHM readily detects lipid droplets during adipocyte differentiation. Because ORO staining is not compatible with DHM acquisition, the lipid bodies in this assay were counter-stained with perilipin 1 (PLIN1, in green), detected with a specific antibody. DAPI was used to stain the nucleoplasm, and PES1 to stain the nucleoli. Small lipid droplets were obvious as early as day 6 of differentiation and increased in size and intensity at later time points. Scale bar, 20 μm. Source data are available online for this figure.

were visible from the onset of differentiation (day 6 and even earlier), becoming larger and more abundant over time to finally merging into supersized structures.

In conclusion, the formation and coalescence of lipid droplets can easily be monitored without staining by use of DHM.

## Discussion

Membraneless organelles, now often referred to as biomolecular condensates, are present in the cell nucleus and cytoplasm, where they play essential roles in modulating all steps of gene expression (Banani et al, 2017). They are formed by liquid-liquid phase separation and are defined by a series of biophysical characteristics that influence their function (Choi et al, 2020). Importantly, it has been shown that the *material state* of condensates can be modulated, with repercussions on the processes that occur inside them. It is therefore essential to develop novel tools in order to probe more deeply the biophysics of condensates. In addition, as

visualization of condensates most often requires specific staining or expression of fluorescently tagged markers that may interfere with their organization and function, for many applications it would be ideal if condensates could also be seen directly with no staining involved.

Here we have applied digital holographic microscopy (DHM, Figs. 1 and EV1), an interferometry-based technique, to detect diverse cell condensates without any staining. Most of the work has focused on the largest nuclear condensate, the nucleolus (Fig. 2), but we have also used the technique to detect smaller condensates in the nucleus and cytoplasm. These include pathological nuclear and cytoplasmic condensates formed upon expression of proteins associated with neurodegeneration (huntingtin, ataxin-3, and TDP-43, Fig. 6) and arsenite-induced cytoplasmic stress granules (Fig. EV4). In all cases, codetection of a specific condensate marker was performed by correlative DHM-fluorescence microscopy. Although DHM may show some limitations in detecting smaller, less dense structures, we have illustrated that pathological condensates are clearly visible and that it is possible to detect at least some SGs. Importantly, we do not imply that

every single SG in a cell can be identified and quantified with this technique.

Fascinating connections are currently emerging between membrane-bound and membraneless organelles. Lipid droplets, which are key to cell metabolism, display features of both types of organelles. We have successfully used DHM to monitor lipid droplet formation without staining in the course of adipose stem cell differentiation to adipocytes (Fig. 7). DHM could thus be a powerful tool for studying the biogenesis and homeostasis of lipid droplets in the future.

As for the nucleolus, we have readily detected it by DHM without staining in a wide diversity of human cells, both adherent and growing in suspension, and notably in cancer cell lines and patient-derived cells (Figs. 2 and EV2). The morphology of the nucleolus could be monitored in unperturbed cells and ones perturbed by drug treatments or factor depletion (Fig. 3). With deep learning and neural network training (Appendix Figs. S1–S3), it was possible to extract automatically the type of information on the nucleolus used in clinical work (e.g., the mean number of nucleoli per cell, mean nucleolar area, etc. Fig. 4). For example, we conclude that in HeLa cells, the mean number of nucleoli per cell is ~2.4, the mean nucleolar surface is ~23 $\mu m^2$, and the mean ratio of the nucleolar-to-nuclear area is ~0.14 (Fig. 4). Note that the extracted features largely correspond to published values (Caragine et al, 2019; Farley et al, 2015; Puck et al, 1956), although in published data the sampling was never as deep as the 3000–6000 nucleoli analyzed here at once. This type of numerical parameter has high potential value in basic research on ribosome biogenesis and in mechanistic studies of processes as essential as tumorigenesis, viral infection, senescence, neurodegeneration, and ageing, among others where nucleolar morphology has been shown to vary greatly (Tiku and Antebi, 2018).

More importantly, DHM is a quantitative technique that captures optical path length variations, which we used to define a novel index: the nucleolar optical thickness. We prove here that the nucleolar optical thickness index can distinguish nucleolar material state changes in several contexts: (1) upon gradual gelation caused by blue light-mediated protein aggregation (Fig. 5), (2) upon drug treatments and factor depletions that disrupt the nucleolar structure (Fig. 3), and (3) upon drug treatment that destabilizes the cytoplasmic cytoskeleton (Fig. EV3). Interestingly, it was possible to observe a dose-response effect, as increased blue light exposure intensity coincided with increased nucleolar gelation (slowed FRAP) and increased nucleolar optical thickness (Fig. 5C–F). Remarkably, it was also possible to group together drugs that similarly impact nucleolar morphology (DRB with ROS and ActD with CX-5461, Fig. 3B). Furthermore, the values of nucleolar optical thickness obtained after depletion of ribosomal protein uL5 or uL18, two proteins that co-assemble on maturing large subunits to form the central protuberance, were surprisingly close (Fig. 3E). We conclude that DHM is particularly suited to probing altered states of cellular condensates.

An alternative to DHM is *Brillouin* microscopy (BM), which probes local spontaneous acoustic wave propagation in viscoelastic samples such as cells (see e.g., (Antonacci and Braakman, 2016). Specifically, BM records the frequency shift of the light scattered inelastically by local spontaneous acoustic waves through the viscoelastic material, providing its high-frequency longitudinal elastic modulus.

As discussed throughout this work, DHM captures optical path length shifts. Thus, while both DHM and BM are non-invasive and require no labeling of the samples, they operate according to different modes of signal capture. Additionally, our DHM plug-in device was simply plugged into a standard optical system, whereas BM relies on more sophisticated custom-built confocal microscopy equipment (Antonacci and Braakman, 2016).

Interestingly, the longitudinal modulus captured by BM has been used as a proxy for cell stiffness (Antonacci and Braakman, 2016). Imaging of a porcine or human cell by BM typically reveals higher longitudinal moduli for the nuclear envelope and nucleoli than for the cytoplasm. The longitudinal modulus appears sensitive to inhibition of actin cytoskeleton formation, values for the cytoplasm being reduced upon latrunculin A treatment, whereas those for the nucleoli appear largely unaffected (Antonacci and Braakman, 2016).

Although we by no means claim that nucleolar optical thickness, as defined here, corresponds to cell stiffness, we thought it would be useful to provide some comparison between DHM and BM. To this end, we also analyzed latrunculin A-treated cells by DHM (Fig. EV3). In keeping with the above BM observations, we observed a reduction in optical thickness in the cytoplasm upon drug treatment. In our case, however, the nucleolus displayed increased optical thickness, which we suggest may result from increased pressure on the nucleolus caused by cell swelling (Fig. EV3). There are thus similarities and differences between the outputs of these two complementary techniques, which was to be expected as they capture different cell characteristics.

Lastly, we show here that DHM is amenable to live-cell imaging, which has allowed us to follow nucleolar genesis and breakdown during mitosis and to monitor dynamically the effects of drugs on nucleolar disruption (Movies EV1,2).

In conclusion, we have developed DHM as a novel tool for qualitatively detecting cell assemblies without staining and for probing the biophysics of condensates. We hope the method will be applied widely in the future to approach fundamental and pathophysiological aspects of organelle biology.

## Methods

### Methods and protocols

#### Cell culture

Cells were grown at 37 °C under 5% $CO_2$. HeLa, HEK293 Flp-In, SiHa, and A549 cells were grown in DMEM (Lonza) supplemented with 10% fetal bovine serum (Sigma), 1% penicillin-streptomycin mix (Lonza). HCT116 and U2OS cells were grown in McCoy's medium (Lonza) supplemented with 10% fetal bovine serum, and 1% penicillin-streptomycin mix. MCF7 cells were grown in EMEM (ATCC) supplemented with 10% fetal bovine serum, human recombinant insulin at 0.01 mg/mL (Sigma), and 1% penicillin-streptomycin mix. LCL, Jurkat, and K562 cells were grown in RPMI (Lonza) supplemented with 15% fetal bovine serum and 2 mM L-glutamine (Lonza). StemPro human adipose-derived stem cells (Invitrogen) were grown in mesenPRO RS Basal medium supplemented with mesenPRO Growth Supplement (Invitrogen) and 2 mM L-glutamine (Invitrogen).

**Reagents and tools table**

| Reagent/Resource | Reference or Source | Identifier or Catalog Number |
|---|---|---|
| **Experimental models** | | |
| HeLa | ATCC | CCL-2 |
| U2OS | ATCC | HTB-96 |
| SiHa | ATCC | HTB-35 |
| HCT116 | ATCC | CCL-247 |
| MCF7 | ATCC | HTB-22 |
| A549 | ATCC | CCL-185 |
| Hela-FBL-GFP | Nicolas et al, 2016; Stamatopoulou et al, 2018 | N/A |
| U2OS G3BP-GFP | Dr Nancy Kedersha (Brigham and Women's Hospital) | N/A |
| LCL #1 | Dr Alyson W. MacInnes (Amsterdam UMC, The Netherlands) | N/A |
| LCL #2 | Dr Alyson W. MacInnes (Amsterdam UMC, The Netherlands) | N/A |
| K562 | ATCC | CCL-243 |
| Jurkat | Pieter Rondou (from Pieter Van Vlierberghe Lab, Ghent University) | N/A |
| HEK293 Flp-In™ T-REx™ cells | Thermo Fisher Scientific | R78007 |
| StemPro human adipose-derived stem cells | Invitrogen | R7788-115 |
| **Recombinant DNA** | | |
| pHR-mCh-Cry2olig | Addgene | 101222 |
| pEGFP-C1-Ataxin3Q84 | Addgene | 22123 |
| pEGFP-Q74 | Addgene | 40262 |
| Tdp43-EGFP construct3 | Addgene | 28196 |
| pcDNA5/FRT/TO | Thermo Fisher Scientific | V652020 |
| pOG44 | Thermo Fisher Scientific | V600520 |
| **Antibodies** | | |
| Rat anti-PES1 | Helmholtz Zentrum München (GmbH) | Clone 8E9 |
| Rabbit anti-RPL11 (polyclonal) | Bethyl | cat# A303-931A |
| Rabbit anti-RPL5 (polyclonal) | Bethyl | cat# A303-933A |
| Mouse anti-beta actin (AC-15) (monoclonal) | Santa Cruz | cat# sc-69879 |
| Goat anti-mouse HRP (polyclonal) | Jackson Immunoresearch | cat#115-035-062 |
| Donkey anti-rabbit HRP | GE Healthcare | cat#NA934v |
| Guinea pig anti-Perilipin 1 (polyclonal) | Progen | cat#GP29 |
| Anti-guinea pig 488 (polyclonal) | Thermo Fisher Scientific | cat#A- 11073 |
| goat anti-rat Alexa Fluo 594 | Thermo Fisher Scientific | A11007 |
| Rat Anti-NST 7H3 | Helmholtz Zentrum Müchen | Clone 7H3 |
| **Oligonucleotides and other sequence-based reagents** | | |
| Scramble siRNA | Thermo Fisher Scientific | 4390843 |
| RPL5 siRNA | Thermo Fisher Scientific | s12153 |
| RPL11 siRNA | Thermo Fisher Scientific | s12170 |
| 5'-GCATCACCACCATCACCATGCCTGCAGGCTCGAGATGGTGTCTAAAGGCGAGG -3' | IDT | Forward mCherry-Cry2olig primer |
| 5'-CGGGCCCTCTAGACTCGATCAGTCACGCATGTTGCAGG-3' | IDT | Reverse mCherry Cry2olig primer |

| Reagent/Resource | Reference or Source | Identifier or Catalog Number |
|---|---|---|
| g-block sequence: GCATCACCACCATCACCATGCCTGCAGGGG AAGATCTGGAAGATCTACAGTGTCCGTATCTAAAAAGGAGAA AAACCGGAAGCGTAGGAACCGAAAGAAGAAGAAAAAGCCCC AGCGGGTGCGAGGGGTGTCCTCTGAGGGTACCCCGATGGAGT CCAACCAAAGCAACAACGGTGGCAGCGGAAACGCCGCGCTTA ACAGAGGAGGCCGCTACGTTCCACCCCATTTGCGGGGTGGCG ACGGTGGGGCTGCAGCCGCTGCTTCTGCAGGCGGGGATGATA GACGAGGAGGCGCAGGAGGTGGAGGTTACCGGCGCGGGGGG GGCAACTCTGGGGGCGGGGGTGGAGGGGGGTTATGATCGCGGC TATAATGACAACCGGGATGACAGGGACAATCGAGGGGGAAGT GGTGGGTATGGTAGGGATAGGAACTACGAGGATAGGGGTTAT AACGGGGGAGGTGGGGGTGGTGGCAATCGGGGCTATAACAAT AATCGAGGTGGGGGAGGTGGAGGTTATAATCGGCAAGATCGG GGGGATGGAGGCTCATCAAATTTTTCCCGAGGTGGATACAATA ATAGGGATGAGGGATCTGACAACCGGGGCAGTGGCCGGTCCTA TAACAACGATAGAAGGGATAACGGCGGTGACGGGCTCGAGAT GGTGTCTAAAGGCGAGGA | IDT | NoLS-RGG |

**Chemicals, Enzymes, and other reagents**

| Reagent/Resource | Reference or Source | Identifier or Catalog Number |
|---|---|---|
| DMEM | Lonza | BE12-604F |
| McCoy's | Lonza | BE12-688F |
| EMEM | ATCC | 30-2003 |
| RPMI | Lonza | BE12-167F |
| MesenPRO RS basal medium + MesenPRO growth supplement | Invitrogen | 12746-012 |
| StemPRO adipocyte differentiation basal medium + StemPRO adipogenesis supplement | Invitrogen | A10070-01 |
| Fetal bovine serum (FBS) | Sigma-Aldrich | F7524 |
| Penicillin-streptomycin | Lonza | DE17-602E |
| L-glutamine | Lonza | BE17-605E |
| Human recombinant insulin | Sigma | I9278 |
| DPBS | Lonza | 17-516 F |
| DAPI | Sigma-Aldrich | D9542 |
| ORO | Sigma-Aldrich | 00625 |
| Methanol | VWR Chemicals BDH | 20903.368 |
| Milk | Nestlé | |
| Tween | Sigma | P9416 |
| Tris | Sigma | T1503 |
| NaCl | Roth | NH00.3 |
| Nitrocellulose blotting membrane | Amersham | 10600007 |
| 30% acrylamide/Bis solution | Bio-RAD | 1610158 |
| SDS | Roth | 2326.5 |
| Ammonium persulfate | Sigma-Aldrich | A3678 |
| TEMED | Roth | 23672 |
| Formaldehyde | Sigma-Aldrich | F8775 |
| Triton X-100 | Sigma-Aldrich | T8787 |
| BSA | Roche | 10735086001 |
| Normal swine serum (NSS) | Vector Laboratories | S-4000-20 |
| MitoTracker GreenFM | Thermo Fisher Scientific | M7514 |
| DMSO | Sigma-Aldrich | D2650 |
| Roscovitine | Sigma-Aldrich | R7772 |
| DRB | Sigma-Aldrich | D1916 |
| Actinomycin-D | Sigma-Aldrich | A1410 |

| Reagent/Resource | Reference or Source | Identifier or Catalog Number |
|---|---|---|
| CX-5461 | Sigma-Aldrich | 5092650001 |
| Doxycycline | Sigma-Aldrich | D9891 |
| Gentamycin | Invitrogen | 15710-064 |
| poly-D-lysine | Thermo Fisher | A38904-01 |
| L-glutamine | Invitrogen | 25030-081 |
| Latrunculin A | Sigma-Aldrich | 428021 |
| Lipofectamine 3000 | Thermo Fisher Scientific | L3000001 |
| InFusion Snap assembly | Takara | 638947 |
| SbfI-HF | NEB | R3642S |
| XhoI | NEB | R0146S |
| Lab-Tek chambered coverglass slides (8 well) | Thermo Fisher Scientific | 155411 |
| Ibidi μ-slide I | Ibidi | 80106 |
| **Software** | | |
| MetaMorph | MDS Analytical Technologies | |
| OsOne | Ovizio | |
| ImageJ | https://imagej.nih.gov/ij/index.html | |
| GraphPad Prism 10 | https://www.graphpad.com | |
| **Other** | | |
| Axio Observer Z1 | Zeiss | N/A |
| pE-2 LED light source | CoolLed | N/A |
| QMOD off-axis differential interferometer | Ovizio | N/A |
| R3 camera | Retiga | N/A |
| Spinning disk confocal head | Yokogawa | N/A |
| Multipoint FRAP module | iLas | N/A |
| CCD camera | HQ2 | N/A |
| Laser bench (405 nm 100 mW Vortran, 491 nm 50 mW Cobolt Calypso, and 561 nm 50 mW Cobolt Jive) | Roper | N/A |
| Single-band bandpass optical filter | Semrock | FF01-550/49-25 |
| 20x (0.5 NA) EC Plan Neofluar | Zeiss | 420350 – 9900 |
| 40x (0.75 NA) EC Plan Neofluar | Zeiss | 420360 – 9900 |
| 63x/1.4 oil DIC Plan-Apochromat | Zeiss | 420782-9900-799 |
| Stage-top incubator system | Live Cell Instruments | |
| Calibration slide | Pyser-SGI | 02A00404 |
| LEDA-B LED array | Teleopto | |
| LAD-1 LED array driver | Teleopto | |

### Cell preparation for DHM imaging

Unless otherwise stated, cells were grown in an Ibidi μ-slide I, fixed in methanol for 5 min at room temperature (RT), washed three times in 1x PBS (Lonza), incubated in DAPI (250 ng/ml, prepared in 1x PBS), and washed three times in 1x PBS for 5 min prior to imaging. For the imaging of suspension cell lines, Ibidi μ-slides were treated with poly-D-lysine (Thermo Fisher) for 15 min and washed 3x with PBS to improve adhesion to the slide. Cells were seeded onto the slides and centrifuged for 5 min at $100 \times g$.

### Correlative DHM-fluorescence

Cells were fixed in 2% formaldehyde (Sigma-Aldrich) for 15 min at RT, washed three times in 1x PBS for 5 min, permeabilized by incubation in 1x PBS /0.3% Triton X-100 (Sigma-Aldrich)/5% BSA (Roche) for 1 h at RT, incubated overnight at 4 °C with the primary antibody diluted 1:1000 in 1x PBS/0.3%Triton X-100/1% BSA, washed three times in 1x PBS for 5 min, incubated for 1 h at RT with the secondary antibody coupled to Alexa dye (Thermo Fisher Scientific) diluted 1:1000 in 1x PBS/0.3% Triton X-100/1% BSA, washed three times in 1x PBS for 5 min, incubated for 15 min at RT with DAPI (250 ng/ml, prepared in PBS 1x), and washed three times in 1x PBS prior to imaging.

### Drug treatment

HeLa cells were grown overnight in Ibidi μslides and treated with the vehicle control (DMSO 0.1%, Sigma-Aldrich), roscovitine (50 μM,

Sigma), DRB (32 μg/mL, Sigma), actinomycin-D (0.2 μg/mL, Sigma), or CX-5461 (5 μM, Sigma-Aldrich) for 2 h prior to DHM imaging. HeLa cells were treated with latrunculin A (500 mM, Sigma) for 30 min.

### siRNA depletion
Cells were transfected with siRNAs as described in (Tafforeau et al, 2013). The siRNAs used (scramble, #4390843, RPL5, #s12153, and RPL11, #s12170) were purchased from Thermo Fisher Scientific.

### Disease condensate induction in cells
Plasmids expressing ataxin-3 with a polyQ expansion (Chai et al, 2002), huntingtin Exon1 with 72 CAG repeats (Narain et al, 1999), and a truncated C-terminal TDP-43 (Yang et al, 2010) were purchased from Addgene. Cells were grown in antibiotic-free complete medium on Ibidi μ-slides and transfected with 0.5 μg of plasmid DNA using lipofectamine 3000 (Thermo Fisher), according to the manufacturer's protocol. Cells were grown for 48 h before fixation and DHM imaging.

### Detection of mitochondria in live-cell imaging
U2OS cells were grown on Ibidi μ-slides for 24 h. Cells were stained with 100 nM MitoTracker FM green (Thermo Fisher) in PBS for 30 min before live-cell imaging.

### Stress granule induction in cells
U2OS G3BP-GFP cells were treated for 1 h with 0.5 mM sodium arsenate dibasic heptahydrate before fixation and DHM imaging.

### Differentiation of human adipose-derived stem cells
To stimulate differentiation, adipose-derived stem cells were seeded in Ibidi μ-slides, and 24 h later, stemPro adipocyte differentiation basal medium (with adipogenesis supplement and gentamycin, Invitrogen) was added. Media was renewed every second day. Lipid droplets were detected either with Oil Red O (Merck) staining, according to the manufacturer's recommendations, or by immunostaining using antibodies specific to perilipin 1.

### DHM image capture
Hardware

Cells were observed by correlative DHM/fluorescence microscopy on a Zeiss Axio Observer Z1 driven by MetaXpress, equipped with LED illumination (CoolLed pE-2), and fitted with a QMOD off-axis differential interferometer (Ovizio s.a.) and a Retiga R3 camera (see Fig. EV1). Holograms were captured using transmitted light (HAL lamp fitted with a single-band bandpass optical filter (Semrock, FF01-550/49-25)) converted with a dedicated software routine. The DHM phase was produced with OsOne. Holograms were imaged with a 20x (0.5 NA) EC Plan Neofluar (Zeiss, 420350 – 9900) or a 40x (0.75 NA) EC Plan Neofluoar (Zeiss, 420360 – 9900) objective and converted to DHM phase with OsOne.

### Pixel-to-size (μm) conversion
Image pixel size was determined as the camera pixel size multiplied by the binning divided by (objective magnification x lens magnification x C-mount). In our setup, these values were: pixel size of the Retiga R3 camera, 4.54 × 4.54 μm; binning, 1x; objective magnification, 20x; lens magnification, 1.5x; C-mount, 1x. For capture at 20x magnification (Fig. 2B and all the images used in the

quantitative analysis), the image pixel size was $(4.54 \, \mu m \times 1)/(20 \times 1.5 \times 1) = 0.151 \, \mu m/pixel$. Note that the pixel-to-size conversion was validated empirically with a calibration slide (Pyser-SGI, 02A00404).

### Image processing pipeline
Description of the database

The database was composed of two datasets of 75 and 50 triplets of images, respectively. A triplet consisted of aligned GFP, DAPI, and DHM images. The resolution of the GFP and DAPI images was $1460 \times 1920$ pixels, and the resolution of the DHM images was $364 \times 480$. The DHM images were interpolated with a spline of order 3 so as to reach the same resolution as the GFP and DAPI images. All images were then cropped to $1408 \times 1920$, as the deep learning model used required every input image dimension to be a multiple of $2^5$. The nucleoli and nuclei were manually contoured on 25 triplets and 10 triplets on the first and second datasets, respectively. The two datasets were captured more than 2 years apart by two different scientists.

### Segmentation of the fluorescence images
Segmentation of nuclei on the DAPI images

The dynamic range of each DAPI image was set at [0, 1] by dividing the intensity of each pixel by its maximum intensity.

The threshold for segmenting DAPI images was set independently on every DAPI image with an automatic approach relying on the fact that the pixel intensity distribution is bimodal in DAPI images. The background and nucleus pixel intensities cluster, respectively, in the first and second modes of the distribution. Every DAPI image was filtered with a Gaussian kernel of standard deviation 3, and the histogram of pixel intensities was computed. Then the threshold was set at the abscissa value where the histogram reached a local minimum between its two modes. This heuristic showed better results than the Otsu algorithm (Otsu, 1979) on the considered dataset. A closing operation (using a disk with radius 3 as a structuring element) was applied in order to fill any small holes occurring within the segmented regions and to obtain a smoother nucleus contour.

As isolated pixels could appear because of thresholding, all regions smaller than 2000 pixels were removed in order to deal with those false detections.

Then a dilation operation with a disk of radius 2 was applied to the binary mask obtained after the thresholding operation. The output of the automatic nucleus segmentation is a black-and-white image where white pixels belong to the nuclei and black pixels to the background (i.e., a binary mask, Appendix Fig. S1A, see inset).

Finally, the binary masks for nucleoli and nuclei were multiplied element-wise in order to remove potential falsely detected nucleoli outside the nuclei in the automatic nucleolus segmentation outputs (see ⊗ symbol in Appendix Fig. S1A).

### Segmentation of the nucleoli on GFP images
The processing pipeline for segmenting the fluorescence images is depicted in Appendix Fig. S1A.

The dynamic range of each GFP image was set at [0, 1] by dividing the intensity of each pixel by its maximum intensity across the dataset. In order to lower the intensity variations across different images of the dataset, the GFP images were further standardized image-wise using the mean and standard deviation of

the pixel intensities belonging to the nuclei in the considered GFP image (using the nuclei segmentation computed on the DAPI images).

Then, nucleoli were identified as pixels whose intensity was above a certain threshold. A multistep approach was necessary because a single threshold did not allow both counting the number of nucleoli per cell nucleus and reliably measuring their area, as a too-low threshold led to fusion of neighboring nucleoli while a too-high threshold led to underestimating the nucleolar area and loss of low-intensity nucleoli.

Therefore, our approach used a sequence of thresholds to first identify all local maxima in the GFP image, after which a region-growing algorithm was used (i) to merge local maxima connected by sufficiently intense pixels and (ii) to connect illuminated pixels to their closest local maximum.

The local maximum detection algorithm considers an increasing sequence of thresholds $[t_0, t_1, \ldots, t_i, \ldots, t_N]$ ranging from $t_0 = 0.083$ to $t_N = 0.3$ with an increment of 0.025. The algorithm iterates from the lowest to the highest threshold. The connected regions segmented with $t_{i+1}$ are always included in the regions obtained with $t_i$. The algorithm allows a region to split into several subregions when the next threshold is considered. However, it does not allow a region to disappear.

After each thresholding operation, two morphological operations are applied: an opening (using a disk with radius 2 as a structuring element) followed by a closing (using a disk with radius 3 as a structuring element). These operations force very close pixels to belong to the same region. When the highest threshold is reached, the seed of each final subregion is defined as its maximum. Hence, there are as many seeds as subregions at the last iteration of the algorithm.

Both region-growing algorithms expand from the computed seeds by adding adjacent pixels until a stopping pixel intensity criterion is met. The region-growing algorithms proposed for nucleolus counting and nucleolar area computation stop when the intensity of the added pixels reaches respectively $\max(\alpha * I_{\text{seed},j}, \tau)$ or $\tau$, where $I_{\text{seed},j}$ is the pixel intensity of seed $j$ and $\tau$ is a constant threshold. Some cells in mitosis (which do not display well-formed nucleoli) are also excluded during this processing step. The GFP signal of those cells has a low intensity and a large area, in contrast to that of cells in interphase, which have either a bright GFP signal or a small area. Cells in mitosis are detected when a region-growing algorithm with a $\tau_{\text{mitosis}}$ stopping threshold provides a region with an area larger than 1000 pixels and a maximum intensity within the region smaller than 0.25. In those cases, the seed is ignored and no region is added. Both region-growing algorithms iterate on all the seeds. In some cases, small holes appear in the grown regions, which are filled. The parameters were tuned to $\alpha = 0.85, \tau = 0.15$ and $\tau_{\text{mitosis}} = 0.11$.

The outputs of automatic nucleolar segmentation are two black-and-white images (one for counting and the other for area computation) where white pixels belong to the nucleoli and black pixels to the background (Appendix Fig. S1A, see insets on the right).

To remove clustered cells and cells whose nucleus touched the border of the images, a postprocessing step was automatically performed. Isolated cells have a convex elliptical nucleus, whereas clustered cells have overlapping nuclei, leading to non-elliptical and concave shapes. The ratio of the area of the considered region to the area of its convex hull was computed. If the ratio is smaller than

0.95, the aggregated nuclei and corresponding nucleoli are removed from the automatic nucleus segmentation output.

### Segmentation of the DHM images

Deep learning architecture

2D U-net belongs to a category of deep learning architectures called convolutional neural networks. These networks are particularly suited for image analysis, as a convolution operation acts as a filter that extracts image patterns. Simple filters can extract edges or corners, but when multiple filters are stacked, complex patterns can be recognized in images. Convolutional neural networks are essentially stacking filters organized in so-called convolutional layers. The 2D U-net model also uses maxpooling layers, in order to merge semantically similar features into one, and rectified linear units (ReLu) as non-linear activation functions (LeCun et al, 2015). The full architecture of the network is shown in Appendix Fig. S1B. The model was trained using Dice loss (Léger et al, 2018). The Adam optimization algorithm was used with a fixed learning rate of $10^{-4}$ (preprint: (Kingma and Ba, 2014). The number of epochs (i.e., the number of cycles through the full training set) was chosen so that convergence was reached. The batch size (i.e., the number of images in every training iteration) was set at one.

For segmentation of the nucleoli, six U-net models were trained in parallel on the same set of training images, but considering different weight initializations. During the testing phase, segmentations were computed with all six models. A pixel was then classified as a part of the nucleolus if at least two predictions labeled it as such. This strategy increases the robustness of the approach. For segmentation of the nuclei, a single U-net model was used.

Nuclei and nucleoli were segmented directly on DHM images by means of deep learning. We used 2D U-net, a state-of-the-art deep-learning architecture for biomedical image segmentation (Falk et al, 2019; Ronneberger et al, 2015). This architecture is a parameterized mathematical function (or model) that maps an input image to a desired output representation with the same size as the input image. In this study, the output representation is an image where the intensity of every pixel is defined between 0 and 1 by a sigmoid function, so that pixels that are close to one, i.e., above 0.5, are classified as a part of a nucleolus, whereas other pixels are classified as background pixels. The same approach is followed for nucleus segmentation. The inner parameters (called weights) of the deep learning model are adapted automatically through a training procedure in order to perform the desired task. This is performed by providing pixel-wise nucleolus or nucleus binary labels to the network in addition to the input images. The training phase proceeds iteratively, by randomly selecting small subsets of samples in the training dataset and adapting the neural network inner weights so that the output predictions fit the desired labels. Once a model is trained, it is used to segment new images, unseen during the training phase. During this testing phase, only input images are provided to the network, which then outputs a binary image with segmentation of the nucleolus or nucleus.

### Learning strategy

The learning strategy is shown in Appendix Fig. S1C. Each DHM image was first normalized by dividing its pixels by their maximal intensity. Then, two separate deep-learning models were trained. One segmented the nucleoli, and the other segmented the nuclei. The nucleolus and nucleus segmentation masks automatically

produced from the fluorescence images were used as labels during the training phase. For nucleolus segmentation, the binary masks obtained for estimating the nucleolar area were used. During the test phase, the processing pipeline analyzed only the DHM images, and the nucleolus and nucleus prediction binary masks were multiplied element-wise in order to remove potential false nucleoli detected outside the nuclei in the automatic nucleolus segmentation output (see $\otimes$ symbol on Appendix Fig. S1C).

We performed threefold cross-validation on the 75 images of the first dataset. This consisted in partitioning the 75 images into three subsets of 25 images. The model was trained on two subsets (i.e., on 50 images), with subset switching in order to test the model on all 75 images. This approach allowed computing the statistics on 75 images without reporting results on the training images. No model retraining was considered for the segmentation of the 50 images of the second dataset. One of the three models used for predictions on the first dataset was arbitrarily chosen.

### Calculation of the nucleolar optical thickness

1. DHM phase images were exported as binary files from OsOne containing the phase shift detected at every (x,y) coordinates, expressed in radians. This phase shift is the product of the physical thickness of the object h(x,y) and its average refraction index n(x,y) at the coordinate (x,y), when the object is immersed in a medium with homogenous refractive index (n0).

$$\text{Phase shift}(x, y) = h(x, y) * (n(x, y) - n0)$$

2. To convert the phase shift into optical path length (OPL), it is multiplied by the light source wavelength $\lambda$ (in nm) and divided by $2\pi$.

$$\text{OPL} = (\text{phase shift} * \lambda)/2\pi$$

3. To determine the optical thickness of the nucleolus, the OPL of the nucleolus was normalized to the OPL of the respective nucleus.

$$\text{Nucleolar optical thickness} = \text{OPL}_{nucleolus} - \text{OPL}_{nucleus}$$

We chose a fixed area of 12 pixels to determine the nucleolar OPL, using the highest intensity values as the center. An equal size area outside the nucleus was used for normalization.

To measure cytoplasmic optical thickness, an area of 12 pixels at approximately 5 µm distance from the nucleus was chosen, and an area of the image containing no cells was used for normalization. The scientist performing the analysis was blinded to the treatment condition of the samples.

### Production of nucleolar Cry2olig cell line

An mCherry-Cry2olig sequence was amplified from plasmid pHR-mCh-Cry2olig (Addgene #101222)(Shin et al, 2017), using 5′-GCATCACCACCATCACCATGCCTGCAGGCTCGAGATGGTG TCTAAAGGCGAGG-3′ and 5′-CGGGCCCTCTAGACTCGATCA GTCACGCATGTTGCAGG-3′ as primers. The PCR fragment was integrated into the pcDNA5/FRT/TO plasmid (Thermo Fisher) by InFusion Snap assembly (Takara). This plasmid was linearized with Sbf I and Xho I, and a synthetic g-block (IDT) containing the NoLS of SF3B2 (Scott et al, 2010), and the RGG domain of LAF1 was

inserted (Elbaum-Garfinkle et al, 2015). HEK293 Flp-In™ T-REx™ cells (Thermo Fisher) were co-transfected with the final plasmid and the pOG44 plasmid (Thermo Fisher), and stable integrated clones were selected with hygromycin.

### FRAP analysis of the nucleolar Cry2olig cell line

Cells were seeded into Lab-Tek chambered coverglass slides (Thermo Fisher). Expression of the construct was induced by incubation with 1 µg/ml doxycycline for 24 h. The cells were kept in the dark after induction of expression. Imaging was performed with a 63x/1.4 oil DIC objective (Plan-Apochromat, Zeiss) on a Zeiss Axio Observer.Z1 microscope driven by MetaMorph (MDS Analytical Technologies, Canada) and equipped with a Yokogawa spinning disk confocal head, an iLas multipoint FRAP module, an HQ2 CCD camera, a laser bench from Roper (405 nm 100 mW Vortran, 491 nm 50 mW Cobolt Calypso and 561 nm 50 mW Cobolt Jive), and a stage-top incubator system (Live Cell Instruments) providing stable 37 °C and 5% $CO_2$. FRAP images were acquired over 5 min in 500 ms intervals. A defined area was bleached by a 98% pulse of the 561 nm laser for 50 ms. Cry2olig oligomerization was achieved by illumination of the cells with the LEDA-B LED array controlled by the LAD-1 LED array driver (Teleopto) at 0 V (No BL), 8 V (Medium BL) or 9.5 V (High BL) for 2 min.

## Experimental design and statistical analysis

The number of independent experiments, n, was: for Figs. 3B and 5F, n = 10; for Fig. 3E, n = 11; and for Fig. EV3, n = 5. Quantifications were performed, when possible, by a scientist blinded to the treatment of the samples. GraphPad Prism 10.1.1 was used for statistical analysis of the data. The statistical tests used are stated in the figure legends and in Expanded View Tables 1–3.

## Data availability

The code was posted in a public repository accessible at https://github.com/legerjean/nucleolus_seg_DHM.

## Peer review information

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

## Acknowledgements

We thank Emilien Nicolas (ULB) for setting up the initial experiments. We thank Laurent Desmecht and André Lebacq (Ovizio s.a) and Gilles Mordant (UCLouvain, statistics helpdesk) for useful advice on data analysis. Research in the Lab of D.L.J.L. was supported by the Belgian Fonds de la Recherche Scientifique (F.R.S./FNRS), BioWin [HOLOCANCER], EOS [CD-INFLADIS], Région Wallonne (SPW EER) Win4SpinOff [RIBOGENESIS], the COST actions EPITRAN (CA16120) and TRANSLACORE (CA21154), the European Joint Program on Rare Diseases (EJP-RD) RiboEurope and DBAGeneCure. Jean Léger (Research Fellow) and Christiane Zorbas (Postdoctoral Researcher) were recipients of F.R.S./FNRS fellowships.

## Author contributions

**Christiane Zorbas**: Conceptualization; Data curation; Formal analysis; Validation; Investigation; Visualization; Methodology; Writing—original draft. **Aynur Soenmez**: Conceptualization; Data curation; Formal analysis; Validation; Investigation; Visualization; Methodology; Writing—original draft. **Jean Léger**: Conceptualization; Data curation; Software; Formal analysis; Investigation; Visualization; Methodology; Writing—original draft. **Christophe De Vleeschouwer**: Conceptualization; Data curation; Software; Formal analysis; Supervision; Validation; Investigation; Visualization; Methodology; Writing—original draft. **Denis LJ Lafontaine**: Conceptualization; Resources; Data curation; Software; Formal analysis; Supervision; Funding acquisition; Validation; Investigation; Visualization; Methodology; Writing—original draft; Project administration; Writing—review and editing.

## Disclosure and competing interests statement

The authors declare that they have no conflict of interest of financial, professional, political, societal, or personal nature with any institute, organization, commercial entity, or person in relation to this work. Although DLJL contributed to the design and solicited the construction of the DHM adapter used in this work, he purchased the finished device from Ovizio s.a. and has no commercial relationship of any sort with Ovizio s.a.

# Expanded View Figures

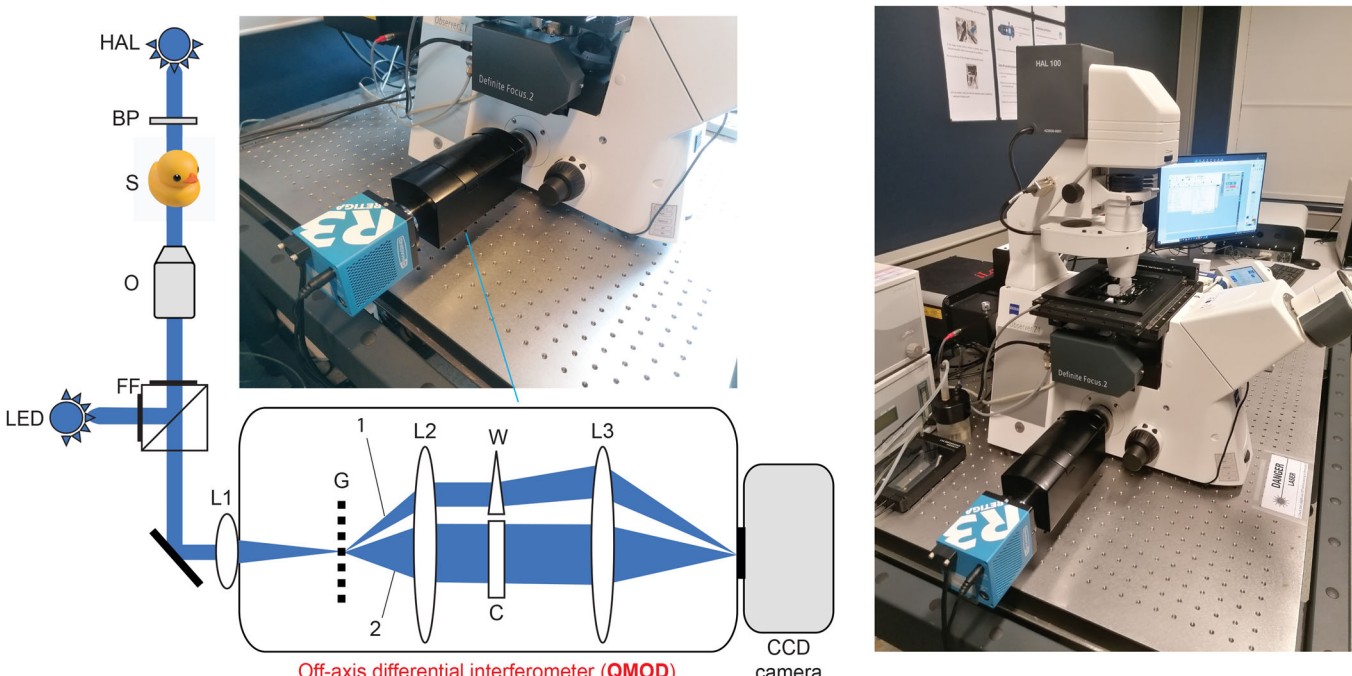

**Figure EV1. Off-axis differential interferometer (QMOD) setup used in this work.**

Description of the beam path and microscope configuration used. The diagram illustrates the detailed beam path in our QMOD setup. A purposely built versatile "plug-in" DHM adapter (QMOD, developed together with Ovizio s.a.) was connected between the lateral port of a Zeiss inverted microscope and a Retiga R3 camera (Qimaging) used for imaging the DHM phase and all fluorescence channels. The camera was driven from the MetaXpress (Molecular Devices) environment. Images were processed with OsOne (Ovizio). Digital holograms were recorded with an incoherent light source (**HAL** lamp) on an inverted microscope adjusted for proper Köhler illumination and coupled to a QMOD interferometer and CCD camera. A bandpass filter (**BP**, 550 nm) was used to increase the coherence of the light and obtain the partial coherence required for holography. The image-forming light rays passing through the specimen (**S**) were captured with the microscope objective (**O**) and directed from microscope lens **L1** to the QMOD interferometer. In the QMOD, a diffraction grating **G** induced splitting of the incident light beam into a diffracted beam (**1**, reference) and a non-diffracted light beam (**2**, object beam). A second lens (**L2**) placed at the focal distance from the grating G reshaped both the diffracted and non-diffracted beams into beams parallel to the optical axis. A wedge (**W**) inserted in the optical path of the object beam induced a slight shift of the images produced by the diffracted and non-diffracted light beams. **C** is a compensating optical module placed in the optical path of the non-diffracted light beam to compensate for the light shift introduced by W in the diffracted beam. The diffracted beam is then recombined with the object beam and focalized by means of objective lens **L3** on the recording plane of a CCD camera, where the hologram is recorded. LED, illumination; FF, fluorescent filter cube. Fluorescence imaging: excitation illumination is emitted by a light-emitting diode (**LED**) and is directed to the sample (**S**) through a fluorescence filter cube (**FF**). Fluorescence emission by the specimen is collected by the objective, passes through the filter cube and **L1**, enters the QMOD, and finally reaches the CCD camera.

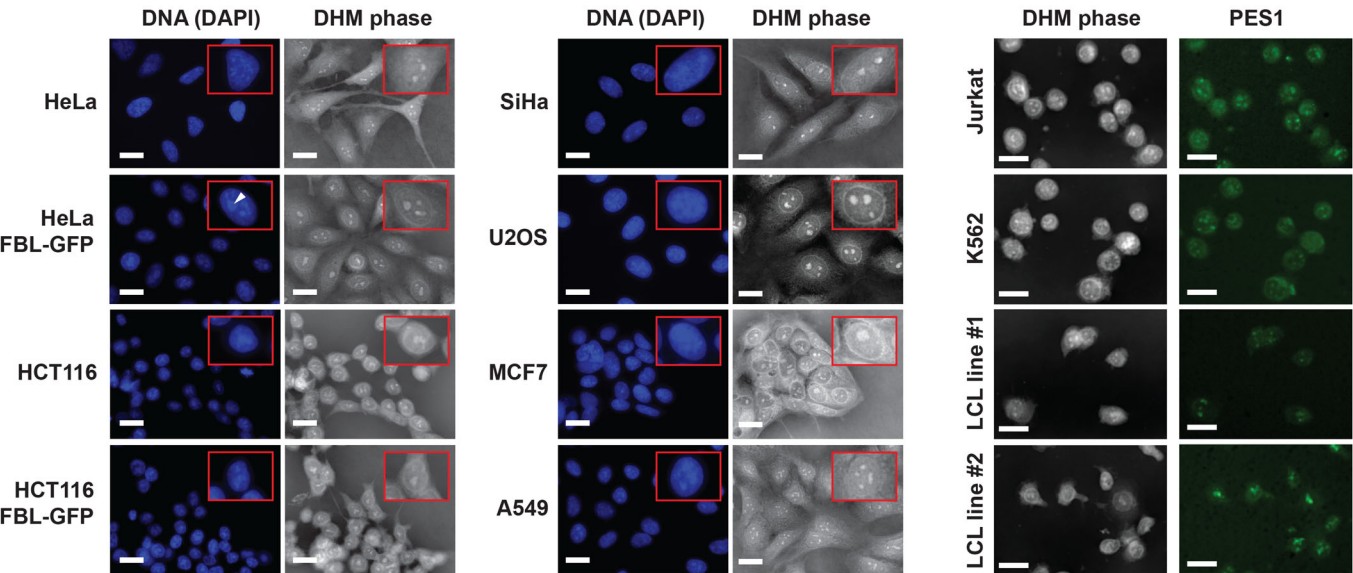

**Figure EV2. DHM detection of the nucleolus in cell lines of various origins.**

Cells stained with DAPI to detect the DNA-rich nucleoplasm were observed by correlative DHM-fluorescence microscopy. (A) Adherent cells: HeLa, HCT116, SiHa, U2OS, MCF7, and A549. Insets, magnification of an individual cell nucleus. An example of a perinucleolar chromatin ring, lining the nucleolus, is highlighted with an arrowhead in the HeLa-FBL-GFP panel for reference (see Fig. 2B for details). Scale bar, 20 μm. (B) Suspension cells: Jurkat, K562, and lymphoblastoid cell lines (LCL). LCL #1 is from a healthy individual; LCL #2 is from a patient expressing a mutation in RPL5. Cells were stained with an antibody against PES1 to detect the nucleolus. Source data are available online for this figure.

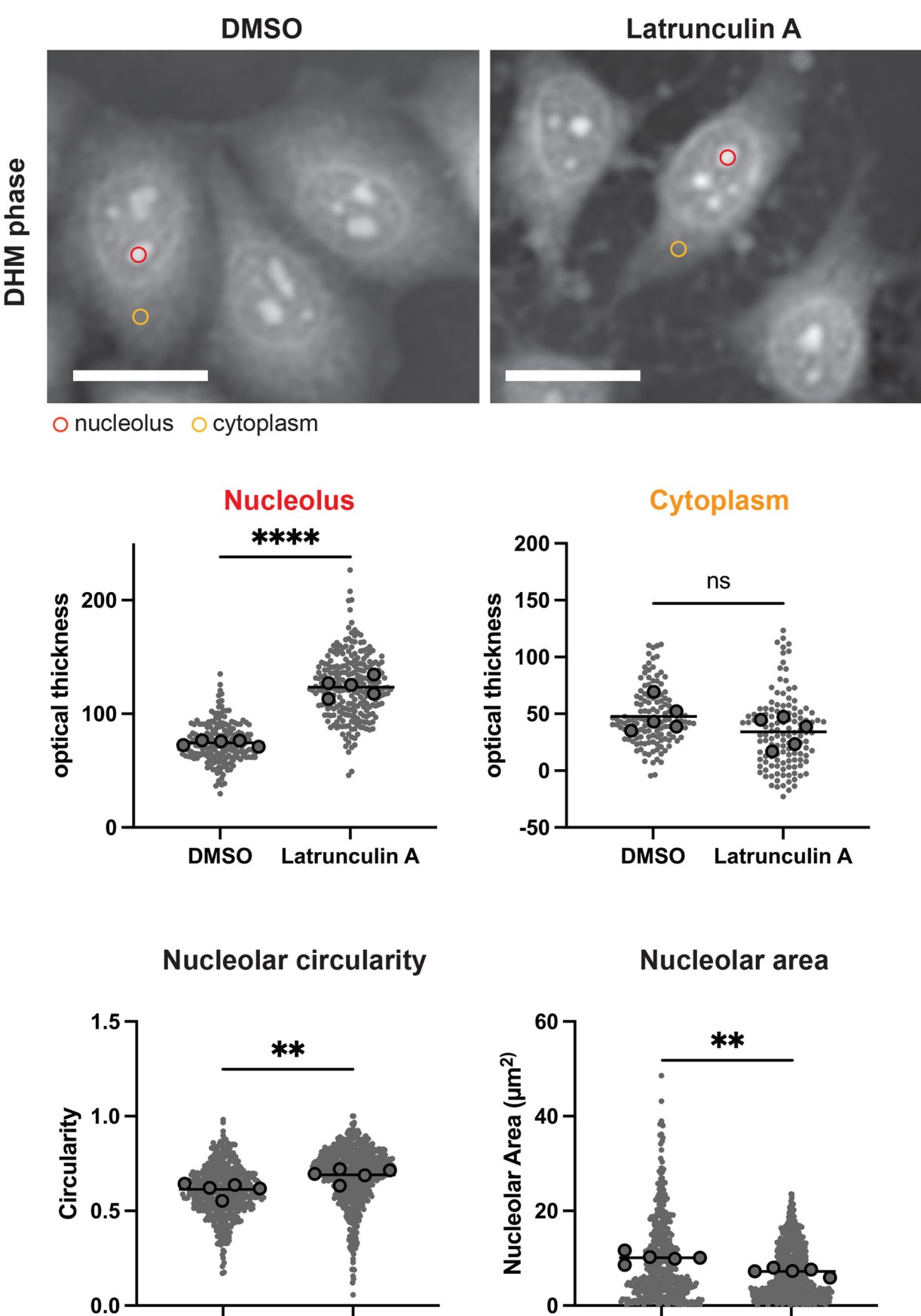

◀  **Figure EV3.   DHM assessment of the effects of latrunculin A on cell material state.**

Quantification of the optical thickness of the nucleolus and cytoplasm in cells treated with the actin cytoskeleton depolymerizing drug latrunculin A (500 mM, 30 min). As control, DMSO was used. Latrunculin A leads to a decrease in cytoplasmic optical thickness and an increase in nucleolar optical thickness. Scale bar, 20 μm. Number of independent experiments, $n = 5$. The mean of each independent experiment is represented by a black circle, the mean of the means by a black line, and the individual nucleoli counted by gray dots. The total numbers of nucleoli counted were 196 and 257 for DMSO and latrunculin A, respectively. The total numbers of cells whose cytoplasm was analyzed were 137 and 129, for DMSO and latrunculin A, respectively. Data were analyzed with the unpaired $t$-test ($p < 0.0001$ for the nucleolus; $p = 0.1535$, for the cytoplasm). Additionally, the nucleolar area and circularity were measured. The nucleolar area is significantly reduced (unpaired $t$-test, $p = 0.0013$). The nucleolar circularity, calculated as $4\pi * \text{area}/\text{perimeter}^2$ is also significantly increased upon latrunculin A treatment (unpaired $t$-test, $p = 0.009$). ns, $p > 0.05$; **$p \le 0.01$; ****$p \le 0.0001$. Source data are available online for this figure.

## arsentite-induced stress granules

### DHM phase  G3BP-GFP

**Figure EV4. Cytoplasmic stress granules can be detected by DHM.**

Stress granule formation was induced in U2OS cells expressing a G3BP-GFP construct by treating them with arsenite (0.5 mM sodium arsenite, 1 h). G3PB detection in the fluorescence channel allowed monitoring the stress granules (red arrowheads), some of which were also visible in the DHM phase. Scale bar, 20 μm. Source data are available online for this figure.

