## [Peer Review File · EMBO Reports]

Detecting Material State Changes in the Nucleolus by Label-free Digital Holographic Microscopy

Denis Lafontaine, Christiane Zorbas, Aynur Soenmez, Jean Léger, and Christophe De Vleeschouwer

Corresponding author(s): Denis Lafontaine (denis.lafontaine@ulb.be)

Review Timeline:

Submission Date:	15th Mar 23
Editorial Decision:	11th May 23
Revision Received:	22nd Dec 23
Editorial Decision:	9th Feb 24
Revision Received:	4th Mar 24
Accepted:	20th Mar 24

Transaction Report:

Dear Prof. Lafontaine

Thank you for the submission of your research manuscript to our journal. I apologize for the delay in handling it but we have now received the full set of referee reports that is copied below.

As you will see, the referees acknowledge that DHM is a potentially useful technique to study nucleolar morphology and material properties. However, they also point out a few technical issues and potential limitations, as for example the application of the technique to study smaller condensates. A formal discussion of the possible application for smaller structures and the resolution limit or the lower density of these condensates must be included in the revised manuscript. Please also address the comment from referee 3 and provide a comparison of DHM to Brillouin microscopy.

Given these constructive comments, we would like to invite you to revise your manuscript with the understanding that the referee concerns (as detailed above and in their reports) must be fully addressed and their suggestions taken on board. Please address all referee concerns in a complete point-by-point response. Acceptance of the manuscript will depend on a positive outcome of a second round of review. It is EMBO reports policy to allow a single round of revision only and acceptance or rejection of the manuscript will therefore depend on the completeness of your responses included in the next, final version of the manuscript.

We realize that it is difficult to revise to a specific deadline. In the interest of protecting the conceptual advance provided by the work, we recommend a revision within 3 months (August 11). Please discuss the revision progress ahead of this time with the editor if you require more time to complete the revisions.

I am also happy to discuss the revision further via e-mail or a video call, if you wish.

We will publish your article in our Resource/Method section. Therefore please check point 8) below.

*******IMPORTANT NOTE:**

We perform an initial quality control of all revised manuscripts before re-review. Your manuscript will FAIL this control and the handling will be DELAYED if the following APPLIES:

- 1) A data availability section providing access to data deposited in public databases is missing. If you have not deposited any data, please add a sentence to the data availability section that explains that.
- 2) Your manuscript contains statistics and error bars based on $n=2$. Please use scatter blots in these cases. No statistics should be calculated if $n=2$.

When submitting your revised manuscript, please carefully review the instructions that follow below. Failure to include requested items will delay the evaluation of your revision. *****

- 1) a .docx formatted version of the manuscript text (including legends for main figures, EV figures and tables). Please make sure that the changes are highlighted to be clearly visible.
- 2) individual production quality figure files as .eps, .tif, .jpg (one file per figure). Please download our Figure Preparation Guidelines (figure preparation pdf) from our Author Guidelines pages <https://www.embopress.org/page/journal/14693178/authorguide> for more info on how to prepare your figures.
- 3) a .docx formatted letter INCLUDING the reviewers' reports and your detailed point-by-point responses to their comments. As part of the EMBO Press transparent editorial process, the point-by-point response is part of the Review Process File (RPF), which will be published alongside your paper.

4) a complete author checklist, which you can download from our author guidelines (). Please insert information in the checklist that is also reflected in the manuscript. The completed author checklist will also be part of the RPF.

5) Please note that all corresponding authors are required to supply an ORCID ID for their name upon submission of a revised manuscript (). Please find instructions on how to link your ORCID ID to your account in our manuscript tracking system in our Author guidelines ()

6) We replaced Supplementary Information with Expanded View (EV) Figures and Tables that are collapsible/expandable online. A maximum of 5 EV Figures can be typeset. EV Figures should be cited as "Figure EV1, Figure EV2" etc... in the text and their respective legends should be included in the main text after the legends of regular figures.

7) Please note that a Data Availability section at the end of Materials and Methods is now mandatory. In case you have no data that requires deposition in a public database, please state so instead of refereeing to the database. See also < <https://www.embopress.org/page/journal/14693178/authorguide#dataavailability>>. Please note that the Data Availability Section is restricted to new primary data that are part of this study.

8) We would kindly ask you to use 'Structured Methods', our new Materials and Methods format, which is mandatory for Method papers (see example:). The Materials and Methods section should include a Reagents and Tools Table (listing key reagents, experimental models, software and relevant equipment and including their sources and relevant identifiers) followed by a Methods and Protocols section in which methods can be described using a step-by-step protocol format with bullet points. More information is available at < <https://www.embopress.org/page/journal/14693178/authorguide#researcharticleguide> >.

9) At EMBO Press we ask authors to provide source data for the main figures. Our source data coordinator will contact you to discuss which figure panels we would need source data for and will also provide you with helpful tips on how to upload and organize the files.

Additional information on source data and instruction on how to label the files are available .

10) The journal requires a statement specifying whether or not authors have competing interests (defined as all potential or actual interests that could be perceived to influence the presentation or interpretation of an article). In case of competing interests, this must be specified in your disclosure statement. Further information: <https://www.embopress.org/competing-interests>

11) Figure legends and data quantification:

- the name of the statistical test used to generate error bars and P values,
- the number (n) of independent experiments (please specify technical or biological replicates) underlying each data point,
- the nature of the bars and error bars (s.d., s.e.m.)

- If the data are obtained from n {less than or equal to} 5, show the individual data points in addition to the SD or SEM.

- If the data are obtained from n {less than or equal to} 2, use scatter blots showing the individual data points.

12) Our journal encourages inclusion of *data citations in the reference list* to directly cite datasets that were re-used and obtained from public databases. Data citations in the article text are distinct from normal bibliographical citations and should directly link to the database records from which the data can be accessed. In the main text, data citations are formatted as

follows: "Data ref: Smith et al, 2001" or "Data ref: NCBI Sequence Read Archive PRJNA342805, 2017". In the Reference list, data citations must be labeled with "[DATASET]". A data reference must provide the database name, accession number/identifiers and a resolvable link to the landing page from which the data can be accessed at the end of the reference. Further instructions are available at .

13) As part of the EMBO publication's Transparent Editorial Process, EMBO reports publishes online a Review Process File to accompany accepted manuscripts. This File will be published in conjunction with your paper and will include the referee reports, your point-by-point response and all pertinent correspondence relating to the manuscript.

Yours sincerely,

Referee #1:

In this work, the authors present the results of studies using digital holographic microscopy to assay changes in the material properties of the nucleolus. This label-free approach has a number of key advantages and opportunities over conventional approaches for characterizing such properties in biomolecular systems and the present work provides excellent use cases. In particular, their use of optical path length (OPL) as a way to segment the nucleus and nucleolus from the cell provides a really intriguing opportunity for future studies. Their use of a correlative approach using fluorescence is quite compelling and well supported. Furthermore, the authors have applied deep learning strategies to help in characterization and interpretation, including image segmentation. The work itself is generally well conducted and clearly described. There are few questions and comments that do bear consideration for clarity.

- How did the authors account for the refractive index of the media in their experiments?
- Based on the DHM images, it would appear that applying an OPL cut-off might suffice to distinguish the various regions. It would be interested to compare a simple cut-off criteria to what the ML model identified.
- Demonstrating the quantitative capabilities of DHM via the change in OPL during the gelation experiment was really nice; however, it would have been nice to correlate this to a change in thickness and / or changes in RI as well.

Referee #2:

In this manuscript, Zorbas et al. described the use of digital holographic microscopy (DHM) to detect and analyze nucleoli in human cells. The application of DHM for label-free imaging of nucleoli avoids overexpression of nucleolar markers which has been highlighted to impede nucleolar organization, e.g. (Jaberi-Lashkari et al. 2022; Riback et al. 2022). The authors utilize an optogenetic system they have published previously (Zhu et al. 2019) and provide conclusive evidence that they can measure quantitative differences in nucleolar optical thickness, thus providing novel information. Detection of nucleoli in adherent cell lines is well described, and multiple cell lines are used to confirm the possibility of DHM, albeit with variable success. Nucleolar form has been well linked to its function and cell health, and thus new techniques for elucidating the changes to nucleolar morphology and material properties upon stimuli and without overexpression will be of interest and timely. Given these obvious strengths, there are a few elements that have lessened my excitement for the manuscript. Notably, the authors either overstate the ease and power of the technique or avoid fully analyzing all the perturbations. Additionally, some controls need to be addressed to confirm that the quantitative measurement of nucleolar optical thickness in DHM is robust to various issues that may impact condition comparisons. I believe these points are likely resolvable without much additional experimentation and

mostly require additional analysis of existing data. The points are detailed below.

Major points:

1. The authors claim that DHM will be useful in detecting nucleoli in both suspension and adherent cells. The only suspension cell line in this study is shown in Figure 2E.
 - a. Nucleoli are not as readily observed in LCL cells when compared to other cell lines. Can the authors include more pictures and/or inset images to clarify this?
 - b. Will robust segmentation of nucleoli be possible for suspension cell lines?
2. Discussion on technical limitations, such as the absence of true 3D imaging with the given DHM setup is missing. Additionally, they should discuss the limitations in axial and lateral resolution with respect to the optical thickness. For example, if nucleoli become smaller in 2D with stimuli will that complicate optical thickness measurements?
3. While exciting in definition, the nucleolar index needs to be characterized better.
 - a. Figure 5F shows a clear difference in nucleolar optical thickness between 2 conditions. However, Figure 5E visually suggests that the optical thickness increases throughout the whole cell upon nucleolar gelation rather than in the nucleolus only. Including a control region outside of the nucleolus (such as the cytoplasm and nucleoplasm) for both conditions is necessary for the data to be convincing.
 - b. The authors performed a variety of drug treatments that affect the nucleolar morphology and/or function shown in Figure 3A. Measurement of the nucleolar index in cells exposed to these drugs would be a good addition to the study. If not possible the limitations should be noted. Controls via analysis on regions outside of the nucleolus as in (a) would likely be needed to ensure robustness to day-to-day changes and/or global cell thickness deviations.
 - c. Have the authors considered other exposures that trigger nucleolar stress, such as heat shock or nutrient deprivation?

Minor points:

1. The authors suggest that DHM is "suitable for studying other biomolecular condensates". Can the authors comment on whether the resolution in DHM is a limiting parameter? Aren't all other biomolecular condensates likely below the resolution capabilities of the DHM setup?
2. What is the average thickness of the nucleus in the selected adherent cell lines? Does it impact the ability of the developed algorithms to quantify the number of nucleoli per cell? Nucleoli are often elongated in human cells, and assuming their number from 2D images can be incorrect.
3. Proper citation of relevant literature is missing in multiple places. For example, 'Typically, numerical parameters such as the mean number of nucleoli per cell nucleus, the mean nucleolar area, and the mean nucleolar-to-nuclear area ratio have proved useful in clinical biology'. Can the authors give examples?
4. Is there a reason why the knockdown of RPLs (Fig 3B) is not confirmed by any of the classical methods used for knockdown validation? Adding a qPCR/WB/immunofluorescence validation is recommended.
5. Description of results:
 - a. In figure legends where tables with numerical data are used, clarify whether +/- indicates SD or another parameter.
 - b. Figure legend for Fig EV3A describes the method, but not the figure. Consider moving the method description to the materials and methods.
 - c. The description of results for Fig EV4A requires clarification, perhaps adding a graphical explanation or a mathematical formula for the calculation of predicted nucleolar area.
 - d. In Figure 4A red and blue lines indicating the segmentation of the nucleoli are too thin. Consider an inset.
 - e. Were technical or biological replicates used to obtain data for Figure 5? Please specify in the figure legends and method section.

Referee #3:

This is a well-written paper where the DHM is exploited to view nucleolus and the method is validated extensively enough by using different perturbations to say that DMH works for viewing nucleoli. This study shows many advantages of DHM technique over traditional microscopy techniques. DHM is a non-invasive technique that does not require staining or labeling of samples. This feature makes it ideal for studying live cells or tissues without altering their natural state. As nucleoli are one of the largest condensates in most human cells, DHM and a similar technique- Brillouin microscopy, which also measures refractive index are able to see nucleoli (10.1038/srep37217). Although this is probably the first paper which does an in-depth analysis of nucleolar structure with DHM. Considering this it may be possible that Brillouin microscopy would be able to do the same. It would have been nice if the author should have discussed Brillouin microscopy and how DHM is better than Brillouin. Authors should discuss if DHM gives a better spatial resolution and should correlate it better to the ultrastructure of the nucleoli. There are also some technical details of DHM which need to be addressed..eg. How does NA and the magnification of the objective used affect the DHM? This will be important to think about if DHM is correlated with the ultrastructure of nucleoli. Also, the paper lacks important explanations about the kind of drug used and how it affects the nucleoli structure and function and how it correlated with DMH images. Further from the measurement of nucleoli, it cannot be generalised that DHM can be used for other

condensates, as other condensates e.g. nuclear speckles and stress granules in the cytoplasm are much less dense than nucleoli. Further, the cytoplasm is rich in condensates, lipid droplets and membrane-bound vesicles. So detailed work will be needed to distinguish between these 3 classes of bodies by DHM.

In my view addressing these issues will increase the quality of the paper and make it suitable for publishing in EMBO.

Major issues-

1. Please elaborate why certain drugs were used, e.g. DRB- what's the function, how it is known to alter nucleoli and how this correlates with DHM of nucleoli.
2. What is the resolution of DHM and how does it change with different magnification and NA of objective? Will DHM resolution change if a different or specific range of wavelength of light is used for holography?
3. The deep learning should have been used to measure intact and fragmented nucleoli and also may be nucleolar area. Can the optical thickness parameter be also used in the AI based learning to identify different nucleolar features?
4. The light intensity or optical activation time can be varied for Cry induced LLPS of nucleoli and then it should be correlated with DHM measurement to see how sensitive is the optical thickness parameter.
5. For dividing cells, DHM does not work very well as seen in Movie 1. This should be mentioned as well. DHM seem to lose resolution in rounded cells. See screen shot..also clear from Movie 1- time 4.04 to 4.52

Minor points

1. All the images lack a scale bar.
2. On page 8- last line of para- "this indicates that the technology is suitable for studying other biomolecular condensates" is too preliminary to say.
3. Insets are not marked with nuclear and cellular boundaries.
4. Figure 4- a graphical representation of the data analysis would be more comprehensive and effective than a tabular representation.
5. The spatial resolution of DHM seems to be lower as compared to fluorescent microscopy, as seen in Figure- 3A, Actinomycin experiment.

In the fluorescent microscopy, three condensates of HeLa-FBL-GFP are visible, whereas DHM shows the presence of one big condensate.

6. Diffused condensates seem to be captured as granular structures with DHM; in Figure 3, DRB experiment, the fluorescent images showed diffused nucleolus, whereas DHM captured the presence of granular nucleolar condensate. How to interpret the different structures?

Point-by-point response to referees comments:**Referee #1:**

In this work, the authors present the results of studies using digital holographic microscopy to assay changes in the material properties of the nucleolus. This label-free approach has a number of key advantages and opportunities over conventional approaches for characterizing such properties in biomolecular systems and the present work provides excellent use cases.

Thank you.

In particular, their use of optical path length (OPL) as a way to segment the nucleus and nucleolus from the cell provides a really intriguing opportunity for future studies. Their use of a correlative approach using fluorescence is quite compelling and well supported.

Thank you.

Furthermore, the authors have applied deep learning strategies to help in characterization and interpretation, including image segmentation. The work itself is generally well conducted and clearly described. There are few questions and comments that do bear consideration for clarity.

- How did the authors account for the refractive index of the media in their experiments?

The *optical thickness* of an object immersed in a homogeneous solution equals its physical height multiplied by its internal refractive index. There is therefore no impact of the refractive index of the medium on our measurements.

We have revised and expanded the corresponding description in the Materials and Methods to make this point clearer. We apologize if there was a confusion on this aspect.

On a somehow related topic, and at the request of Referee #2, we have now systematically subtracted the nucleoplasmic signal from all our nucleolar optical thickness measurements to take background into account.

- Based on the DHM images, it would appear that applying an OPL cut-off might suffice to distinguish the various regions. It would be interested to compare a simple cut-off criteria to what the ML model identified.

Thank you. We understand this suggestion well.

In fact, we initially aimed to do this, but it was not achievable. Applying a simple cut-off (threshold) method to segment the nucleolus on DHM phase captures did not work because of background issues.

In fluorescence mode, the boundary of the nucleolus is very well defined, as can be seen for example in **Figure 2, panel B** (fibrillarin labeling), making a thresholding approach gold standard (the nucleoplasm is 'black').

In the DHM phase captures, it is not so. Instead there is a gradient of pixel intensity across the nucleolar boundary (the nucleoplasm is not 'black'). This reflects the richness of information captured by OPL (each pixel encodes information about the OPL of the objects in the light path), thus preventing a simple thresholding approach on DHM phase images.

- Demonstrating the quantitative capabilities of DHM via the change in OPL during the gelation experiment was really nice; however, it would have been nice to correlate this to a change in thickness and / or changes in RI as well.

We are pleased that Referee #1 acknowledges our demonstration that OPL measurements qualitatively assess cell material states (gelation).

In fact, we have considerably strengthened this conclusion in our revised work, now showing:

(1) that a dose-response can be monitored, the intensity of blue light exposure correlating with the gelation level (assessed by FRAP curves) and nucleolar optical thickness values (assessed by OPL)(see **New Figure 5 panels D and F**).

(2) that, remarkably, drugs known to affect nucleolar morphology similarly can be grouped by nucleolar optical thickness value (DRB with ROS, and ActD with CX-5461)(see **New Figure 3 panel B**).

(3) that depletion of two ribosomal proteins which are normally co-assembled into maturing ribosomes leads to a similar decrease in nucleolar optical thickness (see **New Figure 3 panel E**).

We agree that it would have been ideal to further correlate such changes in OPL with changes in thickness or RI; regrettably this was not achievable with the technology available to us at this stage.

Referee #2:

In this manuscript, Zorbas et al. described the use of digital holographic microscopy (DHM) to detect and analyze nucleoli in human cells. The application of DHM for label-free imaging of nucleoli avoids overexpression of nucleolar markers which has been highlighted to impede nucleolar organization, e.g. (Jaberi-Lashkari et al. 2022; Riback et al. 2022).

Thank you. We completely agree.

The authors utilize an optogenetic system they have published previously (Zhu et al. 2019) and provide conclusive evidence that they can measure quantitative differences in nucleolar optical thickness, thus providing novel information.

Thank you. This is correct. The Cry2tag system has been used before, by us and others.

It may be useful to emphasize, though, that the constructs described in this work (NoLS-LAF1) are completely original.

Detection of nucleoli in adherent cell lines is well described, and multiple cell lines are used to confirm the possibility of DHM, albeit with variable success. Nucleolar form has been well linked to its function and cell health, and thus new techniques for elucidating the changes to nucleolar morphology and material properties upon stimuli and without overexpression will be of interest and timely. Given these obvious strengths, there are a few elements that have lessened my excitement for the manuscript. Notably, the authors either overstate the ease and power of the technique or avoid fully analyzing all the perturbations.

Point well taken, see below.

Additionally, some controls need to be addressed to confirm that the quantitative measurement of nucleolar optical thickness in DHM is robust to various issues that may impact condition comparisons.

Point well taken, see below.

I believe these points are likely resolvable without much additional experimentation and mostly require additional analysis of existing data. The points are detailed below.

Thank you for an overall very positive assessment of our work. Before getting into the specifics, let us provide two general answers:

(1) On the 'ease of use' of the technique:

We would like to emphasize what we believe has been totally overlooked in the assessment of our work, that is: the original development (together with a microscope manufacturer) of a rather unique and simple DHM "plug in" hardware that can be mounted on the C-mount of any inverted microscope, converting it into a DHM platform (see **Figure EV1**).

(2) On the depth of analysis of perturbations:

The work has been substantially strengthened in this direction, and we hope that with the new figures, Referee #2 will now be completely convinced of the potential use and robustness of our approach.

The new figures include:

-**New Figure 3**, presenting a full quantification of alterations after drug treatment and after factor depletion, and the important conclusions drawn from it (drug and factor clustering),

-**New Figure 5**, presenting a dose-response effect analysis of nucleolar gelation.

Major points:

1. The authors claim that DHM will be useful in detecting nucleoli in both suspension and adherent cells. The only suspension cell line in this study is shown in Figure 2E.

a. Nucleoli are not as readily observed in LCL cells when compared to other cell lines. Can the authors include more pictures and/or inset images to clarify this?

Thank you. Initially, we analyzed only one suspension cell line: the patient-derived lymphoblastoid cell line (LCL) presented in **Figure 2E**. Although the nucleolus was detected in these cells by DHM, we agree it was not as easy as in adherent cells.

We now show an analysis of four suspension cell lines (See **New Figure EV2**): LCLs from two individuals, plus Jurkat and K562. We confirm that it was always a bit more difficult to see the nucleolus in suspension cells than in adherent cells, but we clearly show that the nucleoli were visible in all suspension cells tested.

b. Will robust segmentation of nucleoli be possible for suspension cell lines?

Thank you. In principle, detecting nucleoli in suspension cells automatically by AI should be possible, but this would require specific developments (training) that are beyond the scope of the present manuscript. Essentially, this would be another work.

2. Discussion on technical limitations, such as the absence of true 3D imaging with the given DHM setup is missing.

We are not sure we understand exactly what Referee #2 refers to here by ‘true 3-D imaging’? Is it 3-D reconstruction? If it is, our technology does not include 3-D reconstruction (and we never claimed it does). We believe there is a misunderstanding.

The DHM phase images depict the phase shift values in radians. These values are the product of the physical thickness of the object by its refractive index. This implies that phase images depict the whole z-axis information of the object.

Additionally, they should discuss the limitations in axial and lateral resolution with respect to the optical thickness.

The DHM plug-in camera we developed was designed so as to preserve the optical resolution of the microscope platform, which is diffraction limited

In our work, we used 20x and 40x objectives with respective numerical apertures of 0.5 and 0.75, giving resolutions of 671 nm and 447 nm. For conversion of the signal to holograms, we use a specific algorithm (developed by the camera manufacturer) with a routine adapted for 20x and 40x. For our quantifications, we used 20x throughout our manuscript.

See also answers to Referee #3 below.

For example, if nucleoli become smaller in 2D with stimuli will that complicate optical thickness measurements?

Answer: No, absolutely not. This would have no effect, as OPL measurements do not rely on the “2D size” of an object.

3. While exciting in definition, the nucleolar index needs to be characterized better.
a. Figure 5F shows a clear difference in nucleolar optical thickness between 2 conditions. However, Figure 5E visually suggests that the optical thickness increases throughout the whole cell upon nucleolar gelation rather than in the nucleolus only. Including a control region outside of the nucleolus (such as the cytoplasm and nucleoplasm) for both conditions is necessary for the data to be convincing.

In our revised manuscript, we have defined more clearly the *nucleolar optical thickness* in both the main text and the Materials and Methods section, and we have included a correction to take into account the neighboring signal. We believe some of the above questions were raised because the nucleolar optical thickness was not defined well enough, and we sincerely apologize for this.

The *nucleolar optical thickness* is defined as a novel index corresponding to the mean optical path length (OPL) of a 12-pixel-sized disc area centered on the nucleolus in DHM phase images, from which we subtracted the mean OPL of a similar sized area in the adjacent nucleoplasm, to take the background into account (see Materials and Methods for details).

About the need to consider the *neighboring signal*, we absolutely agree. Initially, we did not subtract the nucleoplasmic background from our OPL measurements of the nucleolus. We have now introduced this correction systematically throughout our work. Thank you.

b. The authors performed a variety of drug treatments that affect the nucleolar morphology and/or function shown in Figure 3A. Measurement of the nucleolar index in cells exposed to these drugs would be a good addition to the study. If not possible the limitations should be noted.

We absolutely agree. Thank you.

We have now quantified nucleolar optical thickness systematically in cells after treatment with drugs and after factor depletion (See **New Figure 3 panels B and E**).

The new data interestingly reveal:

(1) that drugs affecting nucleolar morphology similarly group together: DRB with ROS, both of which fragment the nucleolus and lead to reduced OPL values, and ActD with CX-5461, which both lead to formation of clearly identifiable “caps” and also to increased OPL values.

(2) that depletion of ribosomal protein uL5 or uL18 (two proteins which co-assemble to form a remarkable architectural landmark on the large subunit: the central protuberance) leads to similar optical thickness values (reduced values).

Controls via analysis on regions outside of the nucleolus as in (a) would likely be needed to ensure robustness to day-to-day changes and/or global cell thickness deviations.

See Response above. We have now integrated this systematically into our calculation of nucleolar optical thickness.

c. Have the authors considered other exposures that trigger nucleolar stress, such as heat shock or nutrient deprivation?

Yes, we have considered this, but the amount of work involved in this revision was such that it was not possible to perform this also in a timely fashion.

We note that we did trigger nucleolar alterations by several means: drug treatment (4 drugs tested), factor depletion (2 factors depleted), and liquid-to-gel transition (2 blue light exposure conditions). Altogether, we believe we have now accumulated sufficient data to make a clear case for quantitative detection of nucleolar alterations by DHM. We hope Referee #2 will agree with us.

Minor points:

1. The authors suggest that DHM is "suitable for studying other biomolecular condensates". Can the authors comment on whether the resolution in DHM is a limiting parameter? Aren't all other biomolecular condensates likely below the resolution capabilities of the DHM setup?

We have been working with optical resolution since the DHM plug-in camera we developed was installed on a widefield microscope (See original **Figure EV1**).

To explore the potential of this technology to detect both smaller and larger types of cell assemblies in both the nucleus and the cytoplasm, we have detected three types of pathological condensates associated with neurodegeneration (See **New Figure 6**), along with arsenite-induced stress granules (**New Figure EV4**) and lipid droplets (**New Figure 7**). The novel assemblies detected include ones in the nucleus and cytoplasm, cover a wide size range, and include both membraneless and membrane-bound structures.

2. What is the average thickness of the nucleus in the selected adherent cell lines? Does it impact the ability of the developed algorithms to quantify the number of nucleoli per cell? Nucleoli are often elongated in human cells, and assuming their number from 2D images can be incorrect.

-About a potential effect of cell nucleus thickness on the nucleolus count:

The thickness of the nucleus in the inspected cells has no impact on the detection of nucleoli, since we operate in widefield mode and we measure OPL.

-About a potential effect of nucleolar shape on the nucleolus count:

Again, we are in widefield mode. In addition, if there were some bias in counting nucleoli, it would be marginal considering the comprehensive sampling (automatic counting of between 1,200 to 2,000 cells, to produce the data presented in Figure 4).

3. Proper citation of relevant literature is missing in multiple places. For example, 'Typically, numerical parameters such as the mean number of nucleoli per cell nucleus, the mean nucleolar area, and the mean nucleolar-to-nuclear area ratio have proved useful in clinical biology'. Can the authors give examples?

Apologies for this oversight, two key references have been added to support this sentence (Derenzini et al. 2009 and Drygin et al, 2010).

4. Is there a reason why the knockdown of RPLs (Fig 3B) is not confirmed by any of the classical methods used for knockdown validation? Adding a qPCR/WB/immunofluorescence validation is recommended.

Thank you. A western blot analysis proving knockdown efficiency has been added (See **New panel D in Figure 3**).

As noted above, this confirms efficient depletion of uL5 and uL18 and further reveals that these proteins are important for each other's stability (which can easily be rationalized, since they belong with 5S rRNA to the same trimeric complex, and are assembled together in maturing 60 subunits to form the central protuberance).

5. Description of results:

a. In figure legends where tables with numerical data are used, clarify whether +- indicates SD or another parameter.

Thank you. As originally mentioned in our legends, yes, it is standard deviation.

b. Figure legend for Fig EV3A describes the method, but not the figure. Consider moving the method description to the materials and methods.

Figure EV3 has been moved out and is now Appendix Figure S1.

c. The description of results for Fig EV4A requires clarification, perhaps adding a graphical explanation or a mathematical formula for the calculation of predicted nucleolar area.

Figure EV4A is now Appendix Figure S2A.

In fact, these data were already fully described in the main section of our original manuscript, including a mathematical formula. This is basic mathematics, and we believe Referee #2 has overlooked this description.

The original description read (we only updated the Figure number):

... "Interestingly, the mean nucleolar area was reasonably well conserved in nuclei containing up to four nucleoli (**Appendix Figure S2**), after which it gradually increased. This was as expected if small nucleoli coalesce into larger ones in liquid-liquid like fashion, in agreement with the LLPS model of nucleolar assembly. If one views the nucleolus as a sphere, the projected areas of multiple small spheres cover a larger area than the projection of fewer large spheres. Assuming that a constant volume V is divided into N identical spheres, the radius of each sphere becomes proportional to $(V/N)^{1/3}$. Hence, the surface projected by N spheres is proportional to N times $(V/N)^{2/3}$, which is proportional to $N^{1/3}$ (**Appendix Figure S2A**)." ...

d. In Figure 4A red and blue lines indicating the segmentation of the nucleoli are too thin. Consider an inset.

We have increased the thickness of the colored lines.

e. Were technical or biological replicates used to obtain data for Figure 5? Please specify in the figure legends and method section.

The data presented in revised Figure 5 is completely new, since we have now used two illumination conditions side by side (medium and high blue light intensity).

The experiment presented in Figure 5 was repeated multiple times. The data presented correspond to individual cells grown and treated together on the same microchannel slide. We have analyzed >300 nucleoli in each condition.

Referee #3:

This is a well-written paper where the DHM is exploited to view nucleolus and the method is validated extensively enough by using different perturbations to say that DMH works for viewing nucleoli.

Thank you.

This study shows many advantages of DHM technique over traditional microscopy techniques. DHM is a non-invasive technique that does not require staining or labeling of samples. This feature makes it ideal for studying live cells or tissues without altering their natural state. As nucleoli are one of the largest condensates in most human cells, DHM and a similar technique- Brillouin microscopy, which also measures refractive index are able to see nucleoli (10.1038/srep37217). Although this is probably the first paper which does an in-depth analysis of nucleolar structure with DHM. Considering this it may be possible that Brillouin microscopy would be able to do the same. It would have been nice if the author should have discussed Brillouin microscopy and how DHM is better than Brillouin.

Thank you for a positive assessment of work.

Thank you also for pointing out to us Brillouin microscopy, which we had never heard about before.

We are not equipped for Brillouin microscopy, so we cannot provide a direct comparison between the two techniques or comment at this time on whether this technique is better or not than DHM.

Nonetheless, we now discuss Brillouin microscopy in our manuscript as an alternative non-invasive technique for approaching the nucleolus and have added the reference by Antonacci and Braakman (Sci Rep, 2016). We explain in detail the few similarities (non-invasive) and the major differences (signal capture modes and widefield vs. confocal) between the two technologies.

Authors should discuss if DHM gives a better spatial resolution and should correlate it better to the ultrastructure of the nucleoli.

On spatial resolution: We are limited by the optical resolution of our system (diffraction-limited and affected by numerical aperture (NA) of the objectives and the wavelength of light used, see detailed description below).

On correlation to ultra-structure/i.e. electron microscopy (EM): Regretfully, it isn't possible to correlate DHM with EM (nm range), since we are limited by the optical resolution of our system.

There are also some technical details of DHM which need to be addressed..eg. How does NA and the magnification of the objective used affect the DHM?

Point well taken, see below for detailed answer.

This will be important to think about if DHM is correlated with the ultrastructure of nucleoli.

Regretfully, by design, DHM can absolutely not access the ultrastructure of the nucleoli.

Also, the paper lacks important explanations about the kind of drug used and how it affects the nucleoli structure and function and how it correlated with DMH images.

Thank you. We have now provided a full description of how the drugs used (ActD, CX-5461, DRB, and ROS) operate, including necessary references.

Further from the measurement of nucleoli, it cannot be generalised that DHM can be used for other condensates, as other condensates e.g. nuclear speckles and stress granules in the cytoplasm are much less dense than nucleoli.

Although, in our original submission, we only suggested DHM might detect other condensates, we have now explored this further, and we hope Referee #3 will be fully satisfied by the important additional work (**New Figure 6, New Figure 7, New Figure EV4**).

As mentioned above, we have now detected by DHM a range of additional cell assemblies of different sizes, both in the nucleus and the cytoplasm.

These include three types of membraneless disease condensates associated with neurodegeneration, formed upon expression of a mutant form of huntingtin, ataxin-3, or TDP-43 (**New Figure 6**). We have also detected arsenite-induced stress granules (**New Figure EV4**). Lastly, we have monitored membrane-bound assemblies, particularly the biogenesis of lipid droplets formed in the course of stem cell differentiation (See **New Figure 7**). We also detected mitochondria (See **New Movie EV3**).

Further, the cytoplasm is rich in condensates, lipid droplets and membrane-bound vesicles. So detailed work will be needed to distinguish between these 3 classes of bodies by DHM.

Thank you. We now present beautiful detection of cytoplasmic condensates (disease- and stress-induced, **New Figure 6** and **New Figure EV4**) along with lipid droplet condensation (**New Figure 7**), proving it is possible to detect these structures by DHM.

In my view addressing these issues will increase the quality of the paper and make it suitable for publishing in EMBO.

Thank you very much.

Major issues

1. Please elaborate why certain drugs were used, e.g. DRB- what's the function, how it is known to alter nucleoli and how this correlates with DHM of nucleoli.

A full description on how the drugs used operate, with suitable references, has been added.

The new section reads as follows:

... “DRB and roscovitine, which are both Cdk-activated kinase inhibitors, are well known to unfold the nucleolus into beaded strands called nucleolar necklaces (**Fig 3A** and see (Lafontaine *et al.*, 2020; Shav-Tal *et al.*, 2005). They belong to a group of inhibitors of kinases important for transcriptional elongation (Bensaude, 2011). DRB targets CDK9 in P-TEFb and roscovitine inhibits a cdc2-cyclin B kinase that normally keeps RNA Pol I repressed during mitosis (Sirri *et al.*, 2000). Actinomycin-D and CX-5461, on the other hand, are both RNA polymerase I inhibitors, and both of them clearly cause segregation of nucleolar components, which become juxtaposed in so-called “nucleolar caps”, rather than remaining nested (**Fig 3A**, see white arrow and (Lafontaine *et al.*, 2020). ActD is a DNA intercalator which preferentially targets GC-rich sequences and which, at the concentration used here, inhibits only Pol I (although at higher dosage it can inhibit any polymerases). CX-5461 reduces the binding of SL1 pre-initiation complex and RNA polymerase I complex to rDNA promoters (Bywater *et al.*, 2012; Drygin *et al.*, 2011) and stabilizes G-quadruplexes abundant in rDNA (Xu *et al.*, 2017).” ...

2. What is the resolution of DHM and how does it change with different magnification and NA of objective ? Will DHM resolution change if a different or specific range of wavelength of light is used for holography?

About the effects of NA on resolution:

Our DHM plug-in camera module was built to preserve the resolution of our optical system. Of course, as in any diffraction-limited system, increasing the numerical aperture (NA) of the objective will give a better resolution, according to the equation:

$$\text{Resolution} = 0.61 * \text{wavelength (nm)} / \text{numerical aperture of the objective}$$

In our work, we used 20x and 40x objectives with a respective numerical aperture of 0.5 and 0.75, giving resolutions of 671 nm and 447 nm.

For conversion of the signal into holograms, we use a specific algorithm (developed by the camera manufacturer) with a routine adapted for 20x and 40x.

For our quantifications, we used 20x throughout our manuscript.

About the effect of wavelength on resolution:

The light source wavelength and the resolution are inversely correlated: an increase in wavelength has a negative impact on resolution (see above equation).

In our work, the illumination source was a HAL lamp combined with a bandpass filter at 550 nm.

3. The deep learning should have been used to measure intact and fragmented nucleoli and also may be nucleolar area.

Thank you. We have used deep learning to measure nucleolar area (see original **Figure 4**). It was not possible, in the framework of the current study, to use DL to measure fragmented nucleoli, simply because this would imply training entirely new neuronal networks. In essence, this would be another work.

Can the optical thickness parameter be also used in the AI based learning to identify different nucleolar features?

Using AI-based training on OPL captures to identify several nucleolar features is precisely what we did in our work (see original **Figure 4**). These features include: number of nucleoli per cell, nucleolar area per cell, nucleolar diameter, and nucleolar/nuclear area.

If Referee #3 means using AI-based training of OPL captures to detect internal nucleolar features, then there is a need to give an introduction to our answer:

Background to answer: The nucleolus is a multi-layered biomolecular condensate formed by three main layers (FC, DFC, and GC), visible by electron microscopy (nm range) on the basis of their differential electron density or by fluorescence microscopy with suitable staining, on the basis of differential composition. Additional phases, such as the recently described PDFC, are visible only by use of super resolution in combination with specific fluorescence staining.

Answer to the specific question: Our DHM camera was installed on an optical system, and we proved it can detect nucleoli in both unperturbed and perturbed cells (and in the revised version of our manuscript we show it can detect other cell assemblies as well) without staining.

We have detected the nucleolus without staining and we further demonstrate that fine alterations induced by drug treatment or factor depletion can be monitored. In the revised version we show they can be precisely quantified by OPL measurements.

Thus, we show that overall features such as size and shape and the number of nucleoli per cell can be automatically assessed by DL using DHM. We further show that a material state change (caused by loss of function/ribosome biogenesis inhibition and gelation) can be monitored by OPL measurement.

At this stage, it might in principle be possible to monitor internal layers/subphases (i.e. FC, DFC and GC) of the nucleolus by DHM, but this would require extensive experimentation far beyond the scope of the present work. We really appreciate the suggestion. Thank you.

4. The light intensity or optical activation time can be varied for Cry induced LLPS of nucleoli and then it should be correlated with DHM measurement to see how sensitive is the optical thickness parameter.

To address this point directly, we have repeated our 'material state change' experiment presented in **Figure 5**, now using two different blue light illumination conditions.

Specifically, we exposed cells expressing the NoLS-LAF1-mCherry-Cry2olig construct, which self-aggregates upon blue light exposure, to blue light at two different intensities: medium and high (8V and 9.5V, respectively, See **NEW Figure 5**).

We have indeed observed that with increased blue light illumination (High BL) the gelation of the nucleolus was increased, as judged by slower-recovery FRAP curves (compare the green trace, corresponding to high blue light intensity, with the red trace, corresponding to medium-intensity blue light exposure, in **NEW Figure 5 panel D**), and that this correlates with an increase in nucleolar optical thickness, as monitored by OPL measurements (See **NEW Figure 5 panel F**). Specifically, exposure to medium-intensity blue light increases nucleolar optical thickness by 8%, while exposure to high-intensity blue light increases the nucleolar optical thickness by 34%. At least 300 nucleoli were analysed in each condition (See **Figure 5 legends** for detailed statistics).

This new conclusion further demonstrates the potential use of our approach in the booming field of research on biomolecular condensates. Thank you very much for your suggestion, which definitely strengthens our work.

5. For dividing cells, DHM does not work very well as seen in Movie 1. This should be mentioned as well. DHM seem to loose resolution in rounded cells. See screen shot..also clear from Movie 1- time 4.04 to 4.52

Thank you. In fact, the nucleolus is disassembled at the onset of mitosis (coinciding with inactivation of RNA polymerase I through phosphorylation and loss of rRNA synthesis) and it is reassembled at the onset of mitosis (as RNA polymerase I function resumes). Please refer to **Figure 4** in Ref. doi.org/10.1038/s41580-020-0272-6 for a full description of the nucleolar breakdown and nucleolar genesis processes.

Movie EV1 precisely illustrates that loss of detection of the nucleolus during mitosis and its reformation at the end of the process can be seen by DHM.

Minor points

1. All the images lack a scale bar.

Thank you, corrected.

2. On page 8- last line of para- "this indicates that the technology is suitable for studying other biomolecular condensates" is too preliminary to say.

This sentence, as it originally appeared, is no longer part of our revised manuscript.

In the meantime, and as discussed above, we have considerably substantiated our initial suggestion that the technology might be suitable for studying other cell assemblies, with detection of pathological condensates of various sizes in the nucleus and the nucleoplasm (**NEW Figure 6**), of arsenite-induced stress granules in the cytoplasm (**NEW Figure EV4**), and membrane-bound cytoplasmic assemblies of diverse size (lipid droplets biogenesis, **NEW Figure 7**).

3. Insets are not marked with nuclear and cellular boundaries.

We don't understand this comment. To what figure does this comment refer?

4. Figure 4- a graphical representation of the data analysis would be more comprehensive and effective than a tabular representation.

Excellent suggestion. Thank you. We have implemented this change and it is indeed easier to read the data.

5. The spatial resolution of DHM seems to be lower as compared to fluorescent microscopy, as seen in Figure- 3A, Actinomycin experiment. In the fluorescent microscopy, three condensates of HeLa-FBL-GFP are visible, whereas DHM shows the presence of one big condensate.

We understand this comment. It is based on a misunderstanding of the basic organization of the nucleolus and of the staining we used. We apologize for the confusion.

To clarify this point, **Figure 3** has been entirely revised. It presents novel experiments, now with quantification, and also with double staining for fibrillarin (DFC/green) and PES1 (for GC/red).

Answer:

DHM detects the nucleolus 'as a whole', it does not discriminate internal layers/subphases of the nucleolus.

The original FBL staining presented detects specifically an internal layer of the nucleolus (the middle one), so it shows only a 'part of the nucleolus'.

Precisely, actinomycin D treatment leads to segregation of the middle layer, which instead of being "internal" to the nucleolus becomes "peripheral", forming a "Mickey Mouse ears" pattern also termed "caps" (see white arrowheads on **NEW Figure 3 panel A**). A similar pattern is seen upon treatment with CX-5461 (panel underneath in **NEW Figure 3 panel A**), which affects the same step of ribosome biogenesis (the RNA synthesis step -incidentally, we now shown in our revised version that these two compounds lead to a similar increase in nucleolar optical thickness (see panel B). This demonstrates the power of our approach).

Now that we have also labelled the GC layer with PES1, it can clearly be seen that DHM detects the entirety of the nucleolus while the FBL or PES1 labeling detect only specific layers. We hope this description fully answers Referee #3's question.

In summary, the statement that: 'The spatial resolution of DHM seems to be lower than fluorescent microscopy, ...' is incorrect. As discussed above, the resolution of our system is diffraction limited, and the DHM plug-in camera was designed to preserve the resolution of the system.

6. Diffused condensates seem to be captured as granular structures with DHM; in Figure 3, DRB experiment, the fluorescent images showed diffused nucleolus, whereas DHM captured the presence of granular nucleolar condensate. How to interpret the different structures ?

Picture in original submission (single labeling) – no longer part of this manuscript:

This is a continuation of the above comment.

In the new data (double labeling), it can clearly be seen that the DHM and fluorescence captures are completely consistent, provided one keeps in mind that DHM detects all subphases, while the fluorescent staining used here detects one subphase at a time (the DFC with FBL in green and the GC with PES1 in red).

Dear Denis,

Thank you for the submission of your revised manuscript to EMBO reports. We have now received the full set of referee reports that is copied below.

As you will see, the referees consider the revised manuscript much improved, but they raise a number of remaining concerns that need to be addressed. Please address all remaining concerns from referee #2 (regarding statistical analysis based on the number of independent experiments rather than the number of nucleoli). In case the observations are based on the number of nucleoli from one independent experiment, $n = 1$ and all statistical analysis needs to be removed. Please carefully check statistics throughout). Please also address all concerns from referee #3. The potential use and the limitations of DHM to study smaller condensates should be discussed in a transparent manner. I suggest removing the "other cell assemblies (lipid droplets)" from the abstract, if these cannot be reliably identified using DHM.

From the editorial side, there are also a few things that we need:

- Please reduce the number of keywords to 5. I suggest removing 'Cancer', 'Neurodegenerative disease', 'ribosome', and 'cancer'.
- Data availability section: it should only refer to data or code deposited in public repositories. Therefore, please remove 'Data, cells and reagents generated in this study are available upon request.' You also need to remove 'and Material' from the paragraph title.
- The reference 'Kingma DP, Ba J (2014) Adam: A method for stochastic optimization. arXiv preprint: arXiv:1412.6980' is a preprint. The citation in the text should be: (preprint: NAME1 et al, YEAR); in the reference list: Author NAME1, Author NAME2 (YEAR) article title. arXiv doi [PREPRINT].
- Please update the 'Conflict of interest' paragraph to our new 'Disclosure and competing interests statement'. For more information see <https://www.embopress.org/page/journal/14693178/authorguide#conflictsofinterest>
- Please add the following funds, which are listed in the manuscript, in the online submission system: Région Wallonne (SPW EER) Win4SpinOff [RIBOGENESIS], the COST actions EPITRAN (CA16120) and TRANSLACORE (CA21154) and DBAGeneCure
- Please add individual callouts for Figure 1A and 1B, and individual callouts for 3B-E in the text where appropriate.
- Please upload each EV table separately.
- "Extended View Tables" in the text needs to be corrected to "Expanded View Tables" and I suggest adding '1 - 3'.
- Appendix table of content: please add page numbers
- Please remove the movie legends from the manuscript file and provide them in a separate readme.txt file'. Each movie should be zipped with its legend and uploaded as one folder.
- Please insert the 'Reagents and Tools table' in the Methods section. The section is called Materials and Methods, then comes the header 'Reagent and Tools table' with the table, followed by the header 'Methods and Protocols'.
- Source data for main figures need to be uploaded one folder per figure. Source data for EV figures can be grouped into one folder.
- Our production/data editors have asked you to clarify several points in the figure legends (see below). Please incorporate these changes in the manuscript and return the revised file with tracked changes with your final manuscript submission.
 1. Please note that a separate 'Data Information' section is required in the legends of figures 2c-e; EV 2a-b.
 2. Please note that information related to n is missing in the legend of figure 4b.
 3. Although ' n ' is provided, please describe the nature of entity for ' n ' in the legends of figures 3b, e.
 4. Please note that the measure of center for the error bar needs to be defined in the legend of figure 4b.
 5. Please note that the red and white arrowheads are not defined in the legend of figure 6. This needs to be rectified.
- On a different note, I would like to alert you that EMBO Press offers a new format for a video-synopsis of work published with us, which essentially is a short, author-generated film explaining the core findings in hand drawings, and, as we believe, can be

very useful to increase visibility of the work. This has proven to offer a nice opportunity for exposure i.p. for the first author(s) of the study. Please see the following link for representative examples and their integration into the article web page:

<https://www.embopress.org/doi/full/10.15252/emj.2019103932>

With kind regards,

Martina

Referee #1:

The authors have done a terrific job addressing my concerns and those of the other reviewers. The submission is certainly stronger now with the updates and amended text. This is a strong contribution detailed a really nice approach that the authors have developed.

Referee #2:

Overall, the authors have addressed our comments with additional data and analysis. In nearly all places, we find their responses and adjustments adequate. However, we still have substantial concerns with regard to the statistics they use in figures 3B,3E, 5F, EV3 (plots) when evaluating differences between controls and various perturbations (e.g., drugs, siRNA, light, etc). Treating each individual nucleolus as an independent replicate is simply inappropriate especially considering the variances reported. This issue has been discussed at length in the literature (for example, see <https://doi.org/10.1083/jcb.202001064>, notably Figure 1 and Table S1) where using n as the number of observations (in this case nucleoli) instead of the number of experiments (independent applications of the control/treatment) results in an artificially low P-value. We recommend the authors plot the data as in that paper Figure 1B. Furthermore, they should provide the number of independent experiments in their supplemental table.

Referee #3:

After the first set of reviews the authors have done some more experiments, but the the representations and the quantitation of the results are not adequate. The new set of results strongly show that their technique is only suitable for denser condensates like nucleoli and disease related amyloid like aggregates. And hence author should not misled by saying DHM can be used for other cellular assemblies but should say only for denser cellular assemblies.

Major points which needs to addressed before publication:

- 1) Need to compare DHM visibitly of wild type and mutant TDP43 and Ataxin
- 2) TDP43 and Ataxin mutants form foci which is detected by DHM. This also needs quantification, e.g. Fraction of cells which make foci and how many of those foci are visible with DHM.
- 3) SG granule detection efficiency by DHM need quantification.

Specific points to be addressed:

A) Authors write in Abstarct-We conclude that DHM is a powerful tool for quantitatively characterizing nucleoli and other cell assemblies.

I do not agree at all that the DHM is well suited for "physiological" cellular assemblies. As the SGs are not visible at all (please see detailed comment on SG below). Both mutant ataxin and TDP43 -C terminus fragment form denser amyloid like aggregates which are visible in DHM. Authors should have probed and shown wild type Ataxin (known to localize to SG) and wt TDP43

nuclear assemblies, if. They are visible in DHM. Hence they should write- DHM can be used in future to probe altered state of cellular condensates.

B) Authors write in result section-On close inspection, the SGs were also clearly detectable on DHM phase images (Fig EV4). Comment on above- The efficiency of SG detection is very very poor. The authors have just shown one image of a cell where none of the SG which are near the nuclear membrane are visible in DHM. Out of 6 SG in the cells only 1 is clearly visible. 2 of the SGs have actually a kind of depleted region around the SG in DHM, due to which they are visible in DHM. Until, unless authors show a field of cells with stress granules (like how they show for nucleolus) and then show that DHM can capture all the SGs in the field, the authors should not say that the SGs are clearly detectable by DHM. The response to comment how other condensates can be detected by DHM, seems to be done in a hurry and shoddy manner, with no quantifications at all. E.g. how many cells with SG were detected by DHM ? what fractions of SG was detected with DHM.

Zorbas et al. 2024**Editorial requests**

- Please reduce the number of keywords to 5. I suggest removing 'Cancer', 'Neurodegenerative disease', 'ribosome', and 'cancer'.

Done.

- Data availability section: it should only refer to data or code deposited in public repositories. Therefore, please remove 'Data, cells and reagents generated in this study are available upon request.' You also need to remove 'and Material' from the paragraph title.

Done.

- The reference 'Kingma DP, Ba J (2014) Adam: A method for stochastic optimization. arXiv preprint: arXiv:1412.6980' is a preprint. The citation in the text should be: (preprint: NAME1 et al, YEAR); in the reference list: Author NAME1, Author NAME2 (YEAR) article title. arXiv doi [PREPRINT].

Done.

- Please update the 'Conflict of interest' paragraph to our new 'Disclosure and competing interests statement'. For more information see <https://www.embopress.org/page/journal/14693178/authorguide#conflictsofinterest>

Done.

- Please add the following funds, which are listed in the manuscript, in the online submission system: Région Wallonne (SPW EER) Win4SpinOff [RIBOGENESIS], the COST actions EPITRAN (CA16120) and TRANSLACORE (CA21154) and DBAGeneCure

Done.

- Please add individual callouts for Figure 1A and 1B, and individual callouts for 3B-E in the text where appropriate.

Done. Note that the description of 3B and 3E comes later in the text.

- Please upload each EV table separately.

Done.

- "Extended View Tables" in the text needs to be corrected to "Expanded View Tables" and I suggest adding '1 - 3'.

Done.

- Appendix table of content: please add page numbers

Done.

- Please remove the movie legends from the manuscript file and provide them in a separate readme.txt file'. Each movie should be zipped with its legend and uploaded as one folder.

Done.

- Please insert the 'Reagents and Tools table' in the Methods section. The section is called Materials and Methods, then comes the header 'Reagent and Tools table' with the table, followed by the header 'Methods and Protocols'.

Done.

- Source data for main figures need to be uploaded one folder per figure. Sourcedata for EV figures can be grouped into one folder.

Done.

- Our production/data editors have asked you to clarify several points in the figure legends (see below). Please incorporate these changes in the manuscript and return the revised file with tracked changes with your final manuscript submission.

1. Please note that a separate 'Data Information' section is required in the legends of figures 2c-e; EV 2a-b.

Done.

2. Please note that information related to n is missing in the legend of figure 4b.

Done. The number of independent experiments in Fig 4B, n= 2.

3. Although 'n' is provided, please describe the nature of entity for 'n' in the legends of figures 3b, e.

Done. Thanks to referee 2's comments, we have corrected the description of our experiments. In Figure 3B, the number of independent experiments, n= 10. In Figure 3E, the number of independent experiments, n=11.

4. Please note that the measure of center for the error bar needs to be defined in the legend of figure 4b.

Done.

5. Please note that the red and white arrowheads are not defined in the legend of figure 6. This needs to be rectified.

Thank you.

The original arrowheads simply highlighted examples of condensates formed upon expression of the mutant constructs colocalized in the GFP and DHM channels. On the basis of your comment, we realized that we did not assign any particular meaning to the color (red or white) of the arrowheads. This was an oversight.

We now present a revised version of the figure, in which we have assigned a meaning to the color of the arrowheads: red, nuclear condensates; white, cytoplasmic condensates. The figure legend has been corrected.

- On a different note, I would like to alert you that EMBO Press offers a new format for a video-synopsis of work published with us, which essentially is a short, author-generated film explaining the core findings in hand drawings, and, as we believe, can be very useful to increase visibility of the work. This has proven to offer a nice opportunity for exposure i.p. for the first author(s) of the study. Please see the following link for representative examples and their integration into the article web page:

<https://www.embopress.org/doi/full/10.15252/emj.2019103932>

Thank you for this generous offer. Yes, Christiane and Aynur (co-first authors of this work) would be delighted to be involved in the production of such a video-synopsis.

Point-by-point responses to the reviewers' comments

Referee #1

The authors have done a terrific job addressing my concerns and those of the other reviewers. The submission is certainly stronger now with the updates and amended text. This is a strong contribution detailed a really nice approach that the authors have developed.

Thank you very much.

Referee #2

Overall, the authors have addressed our comments with additional data and analysis. In nearly all places, we find their responses and adjustments adequate. However, we still have substantial concerns with regard to the statistics they use in figures 3B,3E, 5F, EV3 (plots) when evaluating differences between controls and various perturbations (e.g., drugs, siRNA, light, etc). Treating each individual nucleolus as an independent replicate is simply inappropriate especially considering the variances reported. This issue has been discussed at length in the literature (for example, see <https://doi.org/10.1083/jcb.202001064>, notably Figure 1 and Table S1) where using n as the number of observations (in this case nucleoli) instead of the number of experiments (independent applications of the control/treatment) results in an artificially low P-value. We recommend the authors plot the data as in that paper Figure 1B. Furthermore, they should provide the number of independent experiments in their supplemental table.

We agree. Thank you very much for this excellent recommendation.

We now present our data as 'independent experiments' rather than simply as counted 'nucleoli'.

In the revised display, the mean of each independent experiment is represented as a 'black circle', the mean of the means as a 'black line', and the individual nucleoli as 'colored dots'. As requested, we now clearly indicate the number of independent experiments in the EV Tables 1-3. For Fig 3B and Fig 5F, n=10; for Fig 3E, n=11; and for Fig EV3, n=5. We have adjusted the figure legends accordingly.

Referee #3

After the first set of reviews the authors have done some more experiments, but the the representations and the quantitation of the results are not adequate. The new set of results strongly show that their technique is only suitable for denser condensates like nucleoli and disease related amyloid like aggregates. And hence author should not misled by saying DHM can be used for other cellular assemblies but should say only for denser cellular assemblies.

We respectfully disagree with the statement that the DHM is suitable only for detecting denser condensates like the nucleoli.

As clearly demonstrated in our work, lipid droplets, which are neither nucleoli nor amyloid-like structures, are nonetheless clearly visible by DHM. In fact, they are so easy to see by DHM that they can be detected even earlier during biogenesis than with classical stains.

Major points which needs to addressed before publication:
1) Need to compare DHM visibilty of wild type and mutant TDP43 and Ataxin

With all due respect, we don't understand this comment. We believe that referee #3 was misled by a clear lack of understanding of the models we have used.

We stress that wild-type TDP43 and ataxin do not form condensates, only the mutant forms do. Therefore this request makes no sense and cannot be addressed experimentally.

For further details, please refer to the following two papers, both of which were already cited in our manuscript:

-for ataxin, doi: 10.1073/pnas.152101299

-for TDP-43, doi: 10.1371/journal.pone.0015878

2) TDP43 and Ataxin mutants form foci which is detected by DHM. This also needs quantification, e.g. Fraction of cells which make foci and how many of those foci are visible with DHM.

At this stage, we wish to stress for referee #3 that this manuscript is focused on the nucleolus. Our aim was truly to develop and apply DHM technology to detect and quantify the nucleolus without staining.

Having done so, we were asked during revision to probe if the technology could also be used to detect smaller and less dense condensates. This is exactly what we did by exploring cell space (cytoplasm vs. nucleus), size range (lipid droplet growth), and disease states.

Specifically, in addition to nucleoli, we have now detected: 1) disease aggregates in both the nucleus and cytoplasm (three types altogether), 2) stress-induced cytoplasmic granules (at least some were detected, see below), 3) mitochondria in live-cell imaging, and finally, 4) lipid droplets in the course of adipose-derived stem cell differentiation to adipocytes.

We hope that in all fairness referee #3 can appreciate that for all those additional bodies we have detected, we did not aim to conduct a systematic quantitative analysis, but a qualitative illustration of the potential of the technique.

We want to stress this again, because each cell assembly would require training a specific algorithm (each structure having distinct morphological and textural features). This is clearly beyond what can be achieved in this work. A full quantitative assessment using DHM is provided for the nucleolus, and we believe that such comprehensive analyses for the other bodies can be left to future works. Thank you.

3) SG granule detection efficiency by DHM need quantification.

In this work we prove it is possible to detect stress-induced cytoplasmic granules (SGs), or at least some of them, by DHM. We never claimed, nor did we intend to claim, that the DHM technique is suitable for detecting every single SG that forms in a cell, in order to achieve a formal and comprehensive quantification.

We have revised our text accordingly:

-Results section:

The former statement reading:

“On close inspection, the SGs were also clearly detectable on DHM phase images (Fig EV4).”

is replaced with the following toned-down claim:

“On close inspection, we realized that at least some SGs could also be detected on DHM phase images (Fig EV4).”

-the Discussion now includes the following new sentences:

“Although DHM may show some limitation in detecting smaller, less dense structures, we have illustrated that pathological condensates are clearly visible and that it is possible to detect at least some SGs. Importantly, we do not imply that every single SG in a cell can be identified and quantified with this technique.”

Specific points to be addressed:

A) Authors write in Abs tarct-We conclude that DHM is a powerful tool for quantitatively characterizing nucleoli and other cell assemblies. I do not agree at all that the DHM is well suited for "physiological" cellular assemblies.

Again, we respectfully disagree with this comment: like the nucleolus, the cytoplasmic lipid droplets are physiological assemblies and they are clearly detected by DHM.

As the SGs are not visible at all (please see detailed comment on SG below).

We have proved that at least some SGs are detectable by DHM. Please see the comment above: we never claimed that every single SG that forms in a cell is detectable/quantifiable by DHM, and SGs were never the focus of this work.

Both mutant ataxin and TDP43 -C terminus fragment form denser amyloid like aggregates which are visible in DHM.

We agree absolutely: we have indeed successfully detected condensates formed upon expression of disease forms of ataxin or TDP-43.

Authors should have probed and shown wild type Ataxin (known to localize to SG) and wt TDP43 nuclear assemblies, if. Hey are visible in DHM. Hence they should write- DHM can be used in future to probe altered state of cellular condensates.

As discussed above, the request to probe wild-type Ataxin and wild-type TDP43 is not reasonable, since they do not form condensates.

Thank you for the suggestion. We have added to our manuscript the notion that DHM is indeed particularly suited to probing altered states of cellular condensates.

The new sentence reads:

“We conclude that DHM is particularly suited to probing altered states of cellular condensates.”

B) Authors write in result section-On close inspection, the SGs were also clearly detectable on DHM phase images (Fig EV4).

Comment on above- The efficiency of SG detection is very very poor.

Thank you.

Initially, we were asked to probe more deeply the space that could be investigated potentially by DHM and, in particular, to show in a revised version that smaller, less dense structures than the nucleolus could be detected. This is exactly what we have done. We never claimed that every single SG in a cell can be detected by DHM.

To take this comment into account, we have toned down our claim on SGs. We now state that “at least some” of them are detectable.

The authors have just shown one image of a cell where none of the SG which are near the nuclear membrane are visible in DHM. Out of 6 SG in the cells only 1 is clearly visible. 2 of the SGs have actually a kind of depleted region around the SG in DHM, due to which they are visible in DHM.

Until, unless authors show a field of cells with stress granules (like how they show for nucleolus) and then show that DHM can capture all the SGs in the field, the authors should not say that the SGs are clearly detectable by DHM.

Thank you. We believe we have understood this repetitive comment on SGs. We have toned down our initial claim, indicating that ‘at least some’ SGs can be detected, and we have removed the ‘clearly’.

The response to comment how other condensates can be detected by DHM, seems to be done in a hurry and shoddy manner, with no quantifications at all. E.g. how many cells with SG were detected by DHM ? what fractions of SG was detected with DHM.

We address above the very repetitive comment on SGs (5 mentions).

We really don't believe it is justified to say our revision was done “in a hurry and shoddy manner”, considering that we have doubled, in the revision, the amount of data.

As mentioned already, we were asked to explore further the 'research space' that could be investigated by DHM and, in particular, to illustrate the potential of the technology to detect smaller, less dense bodies that form in and outside the nucleus. This is exactly what we have done.

We were not asked to perform systematic comprehensive quantitative studies on all the other cell structures that can potentially be detected by DHM, which include at least: pathogenic nuclear and cytoplasmic condensates, mitochondria, lipid droplets, and stress-induced cytoplasmic granules. This would not be achievable in a time frame compatible with publication of this manuscript, as it would require, for a start, developing a specific algorithm for each cell assembly to be characterized.

Prof. Denis Lafontaine
Université libre de Bruxelles
RNA MOLECULAR BIOLOGY
Institut de Biologie et de Médecine Moléculaires
Rue Profs Jeener & Brachet, 12
Charleroi - Gosselies, Hainaut 6041
Belgium

Dear Denis,

I am very pleased to accept your manuscript for publication in the next available issue of EMBO reports. Thank you for your contribution to our journal.

Kind regards,

Martina
